# Diffusion Curriculum: Synthetic-to-Real Data Curriculum via Image-Guided Diffusion

## Abstract

Low-quality or scarce data has posed significant challenges for training deep neural networks in practice. While classical data augmentation cannot contribute very different new data, diffusion models opens up a new door to build self-evolving AI by generating high-quality and diverse synthetic data through text-guided prompts. However, text-only guidance cannot control synthetic images' proximity to the original images, resulting in out-of-distribution data detrimental to the model performance. To overcome the limitation, we study image guidance to achieve a spectrum of interpolations between synthetic and real images. With stronger image guidance, the generated images are similar to the training data but hard to learn. While with weaker image guidance, the synthetic images will be easier for model but contribute to a larger distribution gap with the original data. The generated full spectrum of data enables us to build a novel "**Di**ffusion **CurricuL**um (DisCL)". DisCL adjusts the image guidance level of image synthesis for each training stage: It identifies and focuses on hard samples for the model and assesses the most effective guidance level of synthetic images to improve hard data learning. We apply DisCL to two challenging tasks: long-tail (LT) classification and learning from low-quality data. It focuses on lower-guidance images of high-quality to learn prototypical features as a warm-up of learning higher-guidance images that might be weak on diversity or quality. Extensive experiments showcase a gain of 2.7% and 2.1% in OOD and ID macro-accuracy when applying DisCL to iWildCam dataset. On ImageNet-LT, DisCL improves the base model's tail-class accuracy from 4.4% to 23.64% and leads to a 4.02% improvement in all-class accuracy. [1].

## 1 Introduction

While existing machine learning approaches can train representation or discriminative models with promising generalization performance, their success highly relies on the quality and quantity of the training data. However, in enormous practical scenarios, the data are collected from real environments so neither the quality nor the quantity can always be guaranteed. For example, it is difficult to control the light conditions, weather, motion blur, or the position of objects in the scenes captured by trail/animal cameras, traffic cameras, motion cameras, or robot cameras. Likewise, it is also difficult to keep different classes in the collected data balanced so the model may perform much poorer on tail classes with scarce data. On the other hand, the low-quality/quantity of data also makes the model more prone to the gap between the test and training distributions, thereby posing an out-of-distribution challenge. In many cases, such "hard" training data hinders effective learning, introduces biases or outliers, and may even impact the learning of other data.

Data augmentation and synthesis have been studied to address the challenges of hard real data. By applying pre-defined transformations (Ahn et al., 2023) to data in scarce classes or modifying their backgrounds (Beery et al., 2020; Gao et al., 2022), data augmentation helps learn representations robust to these task-irrelevant variations. While the augmented data may lack sufficient diversity or non-trivial difference to the original data, the recent text-to-image generative models such as GAN or Stable Diffusion enable more sophisticated data synthesis (Dunlap et al., 2024) of diverse higher-quality samples, while the text prompts retain the task-related features. Despite these advancements, existing methods (Han et al., 2024) still struggle to train robust and reliable models or representations for hard classes. Although text-to-image synthesis improves the data quality and quantity, the

---

[1]We opensource the code at link

synthetic data are solely controlled by text prompts but lack sufficient visual similarity to the original image, which leads to a distribution gap to the original data and hurts the generalization performance.

To maximize the merits of synthetic data for learning hard data in real applications and address the syn-to-real gap, we harness the image guidance in diffusion models to generate a full spectrum of interpolations between synthetic data (*i.e.,* generated only from text prompts) and real data (*i.e.,* original images that may suffer from low-quality and low-quantity). The synthetic data at each level of interpolation are generated under the weighted guidance of both the text prompt (e.g., the class name) and the real images. While stronger image guidance preserves visual similarities to the original image, for low-quality or low-quantity data, weaker image guidance could lead to high-quality, diverse, and potentially easier (e.g., with prototypical features) data. Hence, the syn-to-real interpolations create a novel space of synthetic data to design a **generative curriculum** that can adjust the quality, diversity, and/or difficulty of data for different training stages, by selecting the guidance level according to a pre-defined schedule or training dynamics.

In this paper, we develop novel generative curriculum learning approaches for two types of challenging applications with hard real images: long-tail classification, and learning from low-quality images. In *long-tail classification*, learning the tail classes' features is challenging due to their data deficiency and the lack of diversity compared to "head classes". To address this challenge, we propose a curriculum that first learns synthetic images with lower image guidance for tail classes since they enhance the diversity and quantity of the original data. The curriculum then gradually increases the guidance level and learns synthetic images closer to the original images, thereby progressively bridging the syn-to-real gap. In *learning from low-quality data*, the primary challenge is to capture the critical features of the target classes, which is hard due to intricate background, occlusion, or motion blur in the original images. In contrast, images generated with lower image guidance usually contain prototypical features easier to learn. That being said, an overly high or low guidance level may enlarge the domain gap between the training data and the target (in-distribution or out-of-distribution) data. To avoid negative transfer caused by the domain gap and to maximize the merits of synthetic data, we develop an adaptive curriculum that selects the guidance level of synthetic data leading to the greatest progress of each training stage.

We examine two DisCL curricula on benchmark datasets, ImageNet-LT (Liu et al., 2019) and WILD-iWildCam (Beery et al., 2021), for long-tail classification and learning from low-quality images, respectively. Our DisCL curricula improve OOD and ID accuracy by 2.7% and 2.1% respectively on iWildCam. On ImageNet-LT, DisCL improves the minority classes' accuracy by 19.24% and leads to a 4.02% improvement in the overall accuracy. Our main contributions can be summarized as follows:

- We harness image guidance in diffusion models to create a spectrum of synthetic-to-real data for each sample that can be used to design effective training curricula addressing hard data learning.

- We propose the "**Di**ffusion **C**urricu**L**um (DisCL)" paradigm that selects synthetic data of different guidance levels for the needs of each training stage. We propose two novel DisCL curricula to address two important applications, long-tail classification and learning from low-quality data.

- We examine the two DisCL curricula on challenging datasets and demonstrate that DisCL significantly boosts the performance of existing image classifiers especially on the hard data.

## 2  RELATED WORK

**Diffusion models for Synthetic Data**   Recently, a diverse array of generative diffusion models have been proposed, including GLIDE (Halgren et al., 2004), Imagen (Saharia et al., 2022), Stable Diffusion (Rombach et al., 2022), Dall-E (Ramesh et al., 2022), and Muse (Chang et al., 2023). These models can generate realistic, high-resolution images when conditioned on text prompts, and therefore, are used off-the-shelf to augment the datasets for enhancing data diversity. For instance, He et al. (2022) demonstrates that synthetic data created with GLIDE can significantly improve both zero-shot and few-shot performance on image classification. Wu et al. (2023) has explored Stable Diffusion to generate perception data for downstream dense prediction tasks such as Human Pose Estimation, Depth Estimation, and Segmentation. Recent works like Bansal & Grover (2023) and Sariyildiz et al. (2022) have shown that real data combined with synthetic data generated by Stable Diffusion models, boosts the robustness of standard ImageNet classifiers. Other works like Azizi et al. (2023) have finetuned the Imagen model using ImageNet data to enhance the alignment of synthetic data with their classes, while improving the sample diversity. In this work, we utilize

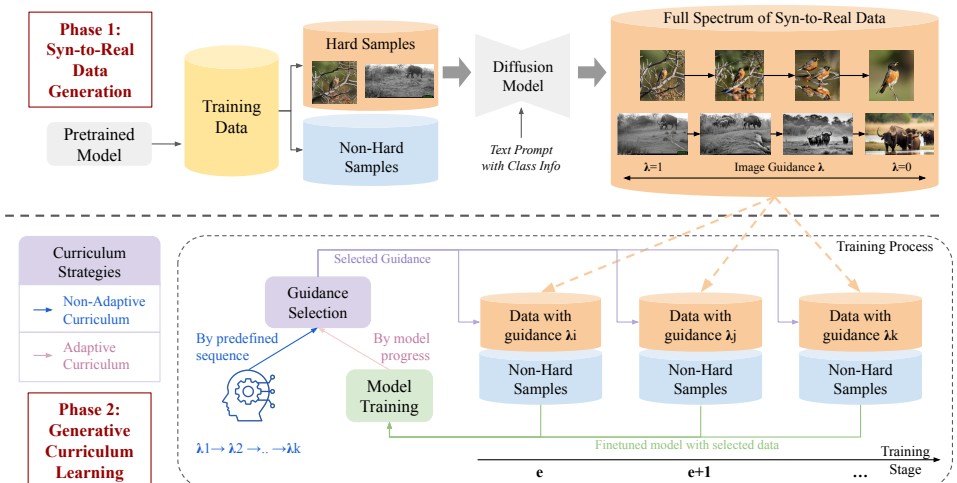

Figure 1: **Overview of Diffusion Curriculum (DisCL)**. DisCL consists of two phases: (Phase 1) Syn-to-Real Data Generation and (Phase 2) Generative Curriculum learning. In Phase 1, we use a pretrained model to identify the "hard" samples in the original images and use them as guidance to generate a full spectrum of synthetic to real images by varying image guidance strength $\lambda$. In Phase 2, a curriculum strategy (Non-Adaptive or Adaptive) selects training data from the full spectrum, by determining the image guidance level for each training stage $e$. Synthetic data of the selected guidance level is then combined with non-hard real samples to train the task model.

off-the-shelf Stable Diffusion models without further finetuning. Unlike previous works, we harness different image guidance levels to generate training images for each stage of model training, thereby progressively learning a full spectrum of interpolations from synthetic to real data.

**Curriculum Learning (CL)**  Curriculum Learning (CL) was first proposed by Bengio et al. (2009), introducing a training method analogous to the step-by-step progressive learning of humans. Subsequent works have further explored this idea; for example, Jiang et al. (2015); Zhou et al. (2020) adjusted the progression pace based on the difficulty of samples, and Jiang et al. (2014); Zhou & Bilmes (2018) further take the data diversity into account. Previous works (Guo et al., 2018; Zhou et al., 2021b; Yuan et al., 2022) have tried CL on more challenging domains like noisy web images and visual QA; this highlights its potential in tackling challenging scenarios. Few works have explored the combination of data augmentation and curriculum learning (Hou et al., 2023), but mainly for the text data (Lu & Lam, 2023; Ye et al., 2021). Some initial efforts have been made by Ahn et al. (2023) to combine CL with engineered image augmentations for tail classes in long-tail learning. In contrast, our work aims to design a generative curriculum on a syn-to-real spectrum of data produced by diffusion models, with broader applications in learning from long-tail or low-quality data.

## 3  METHODOLOGY

We propose diffusion curriculum (DisCL) to *"close the distribution gap between original data and the target data distribution"*. DisCL comprises two phases: (Phase 1) Synthetic-to-Real Data Generation that generates a syn-to-real spectrum of interpolated data for hard samples, and (Phase 2) Generative Curriculum learning based on the synthetic data from Phase 1. The two phases are illustrated in Fig. 1.

### 3.1  SYNTHETIC-TO-REAL DATA GENERATION

**Hard Sample Identification**  We first identify the difficult samples where the model struggles to extract helpful features for target classification. The difficulty estimation can be task-specific. For instance, in long-tail classification with scarce data, the difficulty of each sample depends on whether it belongs to tail classes. For tasks with low-quality data, we can utilize the loss or confidence on the ground-truth class to measure the difficulty. These samples are marked as "hard samples" within the training set (see Fig. 1), to highlight their role in the model's learning process.

**Synthetic Data Generation with Image Guidance**  Classifier-free guidance was initially introduced by Ho & Salimans (2022), to integrate conditional information into the image denoising process

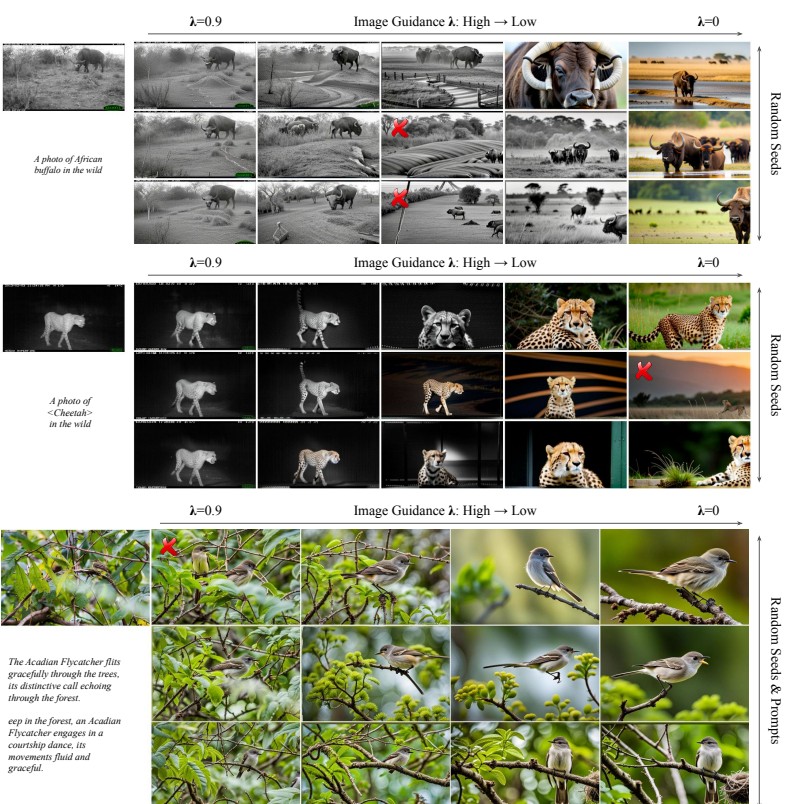

Figure 2: Synthetic images generated with various image guidance levels and random seeds. × marks images with low-fidelity to the text prompt, which are filtered out by CLIPScore (ref. the end of §3.1).

of diffusion models without the need for a classifier. It has been adopted by several Text-to-Image generation models such as Stable Diffusion (SD) (Rombach et al., 2022). Given the original image's latent representation $z_{real}$, the denoising (backward diffusion) process can start from any step $t$ with initial $z_t$ defined as

$$z_t = \sqrt{\tilde{\alpha}_t}z_{real} + \sqrt{1 - \tilde{\alpha}_t}\boldsymbol{\epsilon}, \ \boldsymbol{\epsilon} \sim \mathcal{N}(0, \boldsymbol{I}). \tag{1}$$

The remaining denoising steps iteratively apply the following procedure, noise estimation $\hat{\boldsymbol{\epsilon}}_t$ at each step $t$ and a less noisy generation of $z_{t-1}$, until $t = 0$, resulting in a synthetic image $z_0$.

$$\hat{\boldsymbol{\epsilon}}_t = (1 + w)\boldsymbol{\epsilon}_\theta(z_t, t|c) - w\boldsymbol{\epsilon}_\theta(z_t, t), \ \ z_{t-1} = \frac{1}{\sqrt{\alpha_t}}\left(z_t - \frac{\beta_t}{\sqrt{1 - \tilde{\alpha}_t}}\hat{\boldsymbol{\epsilon}}_t\right) + \sqrt{\beta_t}\boldsymbol{\epsilon}', \ \ t \leftarrow t - 1 \tag{2}$$

In Eq. 1-2, $\tilde{\alpha}_t, \alpha_t$, and $\beta_t$ together define the variance schedule of the diffusion process. $\boldsymbol{\epsilon}, \boldsymbol{\epsilon}' \sim \mathcal{N}(0, \boldsymbol{I})$ are two independently-sampled Gaussian noises, $\boldsymbol{\epsilon}_\theta(\cdot, \cdot)$ refers to the noise estimation model, and $w \in \mathcal{R}$ controls the strength of the textual prompt $c$ as a condition to $\boldsymbol{\epsilon}_\theta(\cdot, \cdot)$.

Since $\tilde{\alpha}_t$ monotonically decreases with $t$, the choice of the initial $t$ in Eq. 1 controls the impact of the original $z_{real}$ in the denoising process, and more visual information of $z_{real}$ tends to be preserved in $z_0$ if initializing from a small $t$. To achieve a full spectrum of interpolations between the real image $z_{real}$ and synthetic images depicted by $c$, we modify the initial step $t$ in Eq. 1 to $t(\lambda) \triangleq \lfloor(1 - \lambda)T\rfloor$ where $\lambda \in [0, 1)$ defines the image-guidance level, i.e.,

$$z_{t(\lambda)} = \sqrt{\tilde{\alpha}_{t(\lambda)}}z_{real} + \sqrt{1 - \tilde{\alpha}_{t(\lambda)}}\boldsymbol{\epsilon}, \ \ t(\lambda) \triangleq \lfloor(1 - \lambda)T\rfloor. \tag{3}$$

Hence, a larger guidance level $\lambda$ leads to higher fidelity of the generated image $z_0$ to the original $z_{real}$, while a smaller $\lambda$ results in a more prototypical image $z_0'$ depicted by textual prompt $c$.

**Synthetic-to-Real Spectrum of Generated Images**   We use state-of-the-art Stable Diffusion Model [2] to generate synthetic images for the hard samples identified in Phase 1 of Fig. 1. By adjusting the

---

[2]We use Stable Diffusion XL model for generation

image guidance scale $\lambda \in [0, 1)$ in Eq. 3, the denoising process in Eq. 2 can produce a full spectrum of smooth transitions between text-only guided synthetic images and real images. We next study the effect of varying the image guidance scales $\lambda$ on the generated synthetic images. As shown in Fig. 2, changing $\lambda$ leads to varying difficulty and diversity of synthetic images. With a smaller $\lambda$, diffusion model mainly relies on the text information provided in the prompt $c$, generating synthetic images that differ markedly from the original and focus more on the distinct prototypical features of the class in $c$. As $\lambda$ increases, the synthetic images increasingly inclines towards the original image, exhibiting less diversity (across random seeds) and more resemblance to the original ones. When the original images are of low-quality, a large $\lambda$ makes it challenging for the classifier to learn discriminating features from synthetic images. Therefore, the broad spectrum of synthetic data offers diverse properties, e.g., diversity, hardness, proximity to the real ones, providing a design space for curriculum learning.

**Filter out Synthetic Data with Low-Fidelity**   As shown in Fig. 2, some synthetic images may suffer from poor quality and low fidelity to the text prompt $c$, *e.g.* the class object is missing or obscured, which are detrimental to the downstream tasks. To mitigate this issue, we perform quality checks and filter out low-fidelity images using CLIPScore (Hessel et al., 2022; Schuhmann et al., 2021), which computes CLIP cosine similarity between synthetic images and the text prompt $c$. We filter out images below some threshold of CLIPScore before using them for training.

### 3.2   Generative Curriculum Learning with Synthetic Data

With the full spectrum of syn-to-real generated data, we achieve a smooth transition from images of prototypical features and high diversity to task-specific features with high resemblance to real images. This enables us to design a curriculum selecting data with according to their diversity and feature types for different training stages. With a curriculum of rich synthetic data, we can enhance the model's performance in challenging and diverse cases which are otherwise difficult to using only the real data. On the other hand, it also allow us to control the distribution gap to the original data.

We apply our method to two challenging applications in the following sections: *long-tail classification* and *learning from low-quality data*. In *long-tail classification*, the scarcity of data in minority/tail classes makes it difficult for models to extract useful features for these classes, leading to poor generalization on balanced test set. To address this, we develop a curriculum strategy that initially exposes the model to diverse synthetic samples of tail classes, and then progressively focuses on task-specific features. This helps mitigate distribution differences between synthetic and real data. In *learning from low-quality data*, the poor quality of data limits the model's ability to detect and extract critical visual features. By employing an adaptive curriculum of synthetic data, we can warm up the model training by learning from varying levels of prototypical features, gradually aiding the model in extracting features useful for out-of-domain generalization.

## 4   Applications

### 4.1   Long-Tail (LT) Classification

**Synthetic Data Generation**   For synthetic data generation, we follow a standard split of tail classes in the studied dataset. Given the real tail-class samples and the associated text prompts, we generate a full spectrum of synthetic data by techniques in §3.1. To mitigate the imbalance among classes, the key is to increase data diversity and quantity for tail classes. We employ a diverse set of textual prompts to achieve the goal[3].

**Generative Curriculum**   The generated spectrum of synthetic data provides varying degrees of data diversity: the images generated with text-only guidance display the highest diversity but may suffer from visual discrepancies to the original images, resulting in a distribution gap that may undermine model performance. To bridge the gap, we progressively shift the synthetic data to a task-specific distribution closer to the original images. This yields a non-adaptive **"Diverse-to-Specific" curriculum** that starts with synthetic data with a lower guidance scale ($\lambda \to 0$) and gradually moves toward data of a higher guidance scale ($\lambda \to 1$).

---

[3]Text prompts are provided in Appendix A.1.2

## 4.2 LEARNING FROM LOW-QUALITY DATA

The data collected in real-world scenarios may suffer from low qualities, such as obscurity in images from traffic, motion, or wildlife observation cameras. We investigate wildlife observation as an example application of DisCL to enable effective learning under such challenging scenarios.

**Synthetic Data Generation**    For low-quality images from camera traps, we aim to generate simpler images containing more prototypical features of the animals that can warm up the training and generalize to more challenging cases. We first identify hard samples based on the ground-truth class probability by a pretrained classifier: a lower probability indicates more difficulty. We vary the image guidance scale to generate a full spectrum of synthetic data for these hard samples, ranging from prototypical to in-the-wild images. The class information is used in the text prompts[4] to steer the diffusion model to generate images relevant to the animals and their wild environment.

**Generative Curriculum**    Training on text-only synthetic data hinders performance due to their distribution gap to the real data and their differences in hardness. A flexible curriculum strategy that integrates both text-only synthetic and real images during training can mitigate this gap. Unlike long-tail classification, various features in the hard samples of low-quality data are not prototypical or generalizable. Synthetic data with higher image guidance can mitigate the issue to some extent but may remain difficult for models to learn from. In contrast, synthetic data with lower image guidance are more prototypical and easier but they are out-of-distribution (OOD) of the real images.

A predefined curriculum of image guidance may introduce OOD features at the early stage, causing a distribution shift, or overemphasizing hard and outlier features, downplaying the prototypical patterns. DoCL (Zhou et al., 2021a) proposed an adaptive curriculum that selects real data for each training stage that can achieve the greatest progress on the original distribution. The curriculum aims to optimize the training dynamics. Inspired by DoCL, we propose an **adaptive curriculum** to dynamically select the guidance level that helps the model achieve the best improvement on the real data distribution. This approach effectively advances the model from learning simple features to mastering more complex and difficult scenarios.

## 5 EXPERIMENTS

### 5.1 LONG-TAIL CLASSIFICATION

**Setup**    To validate the efficacy of DisCL method on long-tail classification, we conduct main experiments with ImageNet-LT (IN-LT) dataset (Liu et al., 2019). This dataset includes 1000 classes, with class cardinality ranging from 5 to 1,280. To assess the robustness of DisCL more comprehensively, we conduct experiments on two additional datasets: a synthetically imbalanced dataset, CIFAR100-LT (Cao et al., 2019), and a real-world benchmark, iNaturalist2018 (Van Horn et al., 2018). CIFAR100-LT is provided with imbalanced classes by synthetically sampling the training data with multiple imbalance ratios $\{100, 50\}$. iNaturalist2018 dataset represents a naturally occurring long-tailed distribution with class cardinality ranging from 2 to 1000. We evaluate overall accuracy and the accuracy across three categories of classes: many (most frequent), medium, and few (least frequent, tail) classes on the standard balanced test sets of three datasets. For synthetic data generation, we use DDIM (Song et al., 2020) as our noise scheduler. For training, following Ahn et al. (2023); Han et al. (2024), we use ResNet-10 as the visual backbone. We average results over 3 runs, with training details in Appendix A.3.1 and hyperparameters in Appendix A.4.

**Baselines**    We compare the effect of DisCL with comparable baseline of CUDA (Ahn et al., 2023) and LDMLR (Han et al., 2024), mainly using Cross-Entropy (CE) loss function. To further illustrate the robustness of DisCL, we try Balanced Softmax (BS) loss (Ren et al., 2020), known for its competitive performance on long-tail learning.

- **CUDA**: Engineered data augmentation + curriculum learning on IN-LT.
- **LDMLR**: A three-stage training using diffusion model to improve LT.
- **BS loss**: Balanced `softmax` to address class-distribution shift between training and test sets.

We also conduct ablation study to analyze the effect of DisCL under different hyperparameter settings. We note that, real data for hard samples ($\lambda \sim 1$) is included by default; however, this doesn't apply to the Fixed Guidance and Text-only Guidance ablation:

---

[4]Text prompts are provided in Appendix A.1.3.

Table 1: Accuracy (%) of long-tail classification on ImageNet-LT with base model ResNet-10. The best accuracy is highlighted in **bold**. † marks our reproduced results using the original paper provided code. BS refers to Balanced Softmax Loss(Ren et al., 2020). Baselines (LDMLR, CUDA) are defined in §5.1.

| | Method | Curriculum | ImageNet-LT | | | |
| | | | Many | Medium | Few | Overall |
|---|---|---|---|---|---|---|
| Baselines | CE | N/A | 57.70 | 26.60 | 4.40 | 35.80 |
| | CE + LDMLR | N/A | 57.20 | 29.20 | 7.30 | 37.20 |
| | CE + CUDA | N/A | 57.49 | 28.16 | 6.58 | 36.30 |
| | BS† | N/A | 51.14 | 37.02 | 19.29 | 39.80 |
| | BS + CUDA† | N/A | 51.16 | 37.35 | 19.28 | 40.03 |
| Ablations | CE + Text-only Guidance | N/A | 56.63 | 30.69 | 17.90 | 39.10 |
| | CE + All-Level Guidance | N/A | 56.77 | 30.88 | 19.17 | 39.40 |
| | CE + DisCL | Adaptive | 56.21 | 30.43 | 16.78 | 38.65 |
| | CE + DisCL | Specific to Diverse | 56.71 | 30.67 | 18.36 | 39.18 |
| | CE + DisCL [Lower CLIPScore Threshold] | Diverse to Specific | 57.66 | 30.61 | 23.69 | 39.67 |
| | CE + DisCL [Higher CLIPScore Threshold] | Diverse to Specific | 56.92 | 30.64 | 22.88 | 39.68 |
| Ours | CE + DisCL | Diverse to Specific | 56.78 | 30.73 | **23.64** | 39.82 |
| | BS + DisCL | Diverse to Specific | 52.68 | **37.68** | 21.36 | **41.33** |

Table 2: Accuracy (%) of long-tail classification on CIFAT-100-LT with base model ResNet-10. The best accuracy for classes of {many, medium, few} samples is highlighted in **bold**. Baselines are defined in §5.1.

| | | CIFAT-100-LT | | | | | | | |
| | | Imbalance Ratio=100 | | | | Imbalance Ratio=50 | | | |
| Method | Curriculum | Many | Medium | Few | Overall | Many | Medium | Few | Overall |
|---|---|---|---|---|---|---|---|---|---|
| CE | N/A | 52.86 | 25.34 | 5.49 | 29.02 | 49.60 | 25.41 | 5.33 | 31.72 |
| CE + CUDA | N/A | 54.55 | 26.07 | 5.43 | 29.02 | 52.29 | 26.17 | 5.53 | 33.13 |
| **CE + DisCL** | Diverse to Specific | 53.14 | 25.52 | **13.65** | **39.91** | 53.4 | 31.69 | **21.47** | **36.22** |
| BS | N/A | 47.87 | 30.07 | 14.41 | 31.61 | 46.01 | 30.76 | 18.55 | 34.82 |
| BS + CUDA | N/A | 48.01 | 32.79 | 15.55 | 33.02 | 46.08 | 32.51 | 22.11 | 36.21 |
| **BS + DisCL** | Diverse to Specific | **49.02** | 29.02 | **19.07** | **33.08** | **49.51** | 32.6 | **25.58** | **36.77** |

- **Text-only Guidance**: Using data at image guidance scale $\lambda = 0$ without curriculum strategy.

- **Fixed Guidance** [5]: uses data generated from a single guidance scale $\lambda_i \in [0, 1)$. We report results for the guidance with the highest few-class accuracy.

- **DisCL**: employs multiple levels of guidance scales alongside a range of curriculum strategies. These strategies and the guidance intervals used for training, are defined below:
  - **Specific to Diverse**: Non-adaptive strategy with guidance changing from largest (task-specific augmentation) to smallest (diverse augmentation).
  - **Diverse to Specific**: Non-adaptive strategy with guidance changing from smallest to largest.
  - **Adaptive**: Curriculum strategy[6] to adaptively select guidance during training.

**Results** We present the results of our method alongside the baselines for the ImageNet-LT dataset in Table 1. With CE loss, DisCL significantly improves accuracy in all 4 class-categories. Notably, "Few" class accuracy increases by 17.06%, from 6.58% to 23.64%, demonstrating DisCL's effectiveness in addressing the data scarcity challenge, especially for tail classes. We also try our DisCL method with BS loss, and observe additional gains (1.52% in Many, 2.08% in Few, and 1.3% Overall); this emphasizes the impact of our approach even with a class-balancing loss function. The results on CIFAR100-LT and iNaturalist2018 (as shown in Table 2 and Table 3) further demonstrate the robustness of DisCL on various datasets. These experimental results shows that by utilizing a diverse spectrum of data, our method achieves better accuracy in tail classes, alongwith improved overall generalization.

## 5.2 LEARNING FROM LOW-QUALITY DATA

**Setup** We also conduct DisCL experiments with iWildCam dataset (Beery et al., 2021) to evaluate its efficacy in classifying low-quality data. The task is to classify 182 different animal species from

---

[5]*Text-only Guidance* ($\lambda$=0) reaches the best performance amongst all guidance scales. Hence, the result of *Fixed Guidance* here are same as *Text-only Guidance*, reported in Table 1. We also report the performance of all other scales in *Fixed Guidance* experiment in the Fig. 12.

[6]Curriculum strategy proposed in §4.2

Table 3: Accuracy (%) of long-tail classification on iNaturalist2018 with base model ResNet-10. The best accuracy is highlighted in **bold**. Baselines are defined in §5.1.

| Method | Curriculum | iNaturalist2018 | | | |
| | | Many | Medium | Few | Overall |
|---|---|---|---|---|---|
| CE | N/A | 55.02 | 43.40 | 37.33 | 42.20 |
| CE + CUDA | N/A | 55.94 | 44.21 | 39.13 | 43.18 |
| **CE + DisCL** | Diverse to Specific | 54.71 | 44.37 | **48.92** | **47.25** |
| BS | N/A | 46.12 | 49.31 | 50.27 | 49.46 |
| BS + CUDA | N/A | 48.77 | 49.94 | 50.87 | 50.23 |
| **BS + DisCL** | Diverse to Specific | 45.44 | 48.18 | **53.63** | **50.30** |

images captured by camera traps. We evaluate model performance on standard out-of-domain (OOD) and in-domain (ID) test sets in terms of macro F1 score. We choose the CLIP ViT model as our base model and finetune CLIP ViT-B/16 and CLIP ViT-L/14 [7] models with DisCL. The reported accuracy is averaged over 3 random seeds. More training details and hyperparameters are provided in Appendix A.3.2 and Appendix A.4.

**Baselines**  We compare the effect of our method with three benchmark algorithms, LP-FT (Kumar et al., 2022), FLYP (Goyal et al., 2023), and ALIA (Dunlap et al., 2024). To further analyze the gain of our model, we try Weighted Ensembling (WE) method (Wortsman et al., 2022), which can further improve model performance by integrating prior knowledge from pretrained model:

- **LP-FT**: A two-step process involving linear probing and full fine-tuning of model to avoid distortion of pretrained features, to improve OOD generalization.
- **FLYP**: Finetuning with the pretraining loss (contrastive loss).
- **ALIA**: Diffusion-based data-augmentation on fine-grained classification tasks.
- **WE**: Linearly merging the weights of pretrained and finetuned model.

We conduct ablation study to analyze the effect of DisCL with different hyper-parameters introduced in §5.1, and the newly introduced ablation hyper-parameters:

- **DisCL**: employs multiple levels of guidance scale and a range of curriculum strategies:
  - **Easy to Hard**: Non-adaptive strategy with guidance changing from smallest (easiest and most prototypical features, $\lambda \sim 0$) to largest (hardest and most task-specific features, $\lambda \sim 1$).
  - **Random**: Randomly selecting guidance at each training stage.

**Results**  We present the results of our method and comparable baselines for the iWildCam dataset in Table 5. Compared to the nearest competing baseline, DisCL significantly enhances the OOD F1 performance by 2.6%. Additionally, DisCL boosts the ID F1 performance by 2.1%. Among all evaluated methods, DisCL achieves the highest scores in both OOD and ID metrics, underscoring its effectiveness for this low-quality classification task. Moreover, our model could still deliver performance improvements on larger model when using ViT-L/14, as shown in Table 4; DisCL achieves gains of 2.8% in OOD F1 and 3.7% in ID F1. These findings reinforce the versatility and robustness of the DisCL framework across different model scales and complexities. We further study the performance of model after employing WE method. DisCL still benefits from this method and maintains superior performance compared to other methodologies, despite integrating prototypical features from synthetic data that might overlap with the pretrained model's knowledge.

Table 4: F1 Score with CLIP ViT-L/14

| Method | iWildCam | | | |
| | Without WE | | With WE | |
| | OOD | ID | OOD | ID |
|---|---|---|---|---|
| CLIP (Zero-Shot) | 12.1 | 11.8 | 12.1 | 11.8 |
| FLYP† | 40.3 | 55.9 | 41.9 | 57.7 |
| **FLYP + DisCL** | **43.1** | **59.6** | **44.8** | **60.2** |

## 6  Ablation Study and Analysis

### 6.1  Effect of Syn-to-Real Interpolation Data

We examine the effectiveness of using a spectrum of data generated with our DisCL method, by comparing *All-Level Guidance* and *Text-only Guidance* rows in both the task tables (IN-LT and

---

[7]We use hyperparameters provided in Goyal et al. (2023) with a batchsize of 128 to train the model.

Table 5: In-distribution (ID) and out-of-distribution (OOD) macro F1 score of low-quality image learning on iWildCam with CLIP ViT-B/16 model. The best performance is highlighted in **bold**. † marks our reproduced results using the original paper provided code. Baselines are defined in §5.2.

| | Method | Curriculum | iWildCam OOD | iWildCam ID |
|---|---|---|---|---|
| Baselines | CLIP (zero-shot) | | 11.0 (-) | 8.7 (-) |
| | LP-FT | N/A | 34.7 (0.4) | 49.7 (0.5) |
| | LP-FT + WE | N/A | 35.7 (0.4) | 50.2 (0.5) |
| | FLYP† | N/A | 35.5 (1.1) | 52.2 (0.6) |
| | FLYP + WE† | N/A | 36.4 (1.2) | 52.0 (1.0) |
| Ablations | FLYP + Text-only Guidance | N/A | 34.2 (0.4) | 51.4 (0.3) |
| | FLYP + Fixed Guidance | N/A | 36.0 (0.3) | 50.8 (0.6) |
| | FLYP + All-Level Guidance | N/A | 36.5 (0.6) | 53.4 (0.5) |
| | FLYP + DisCL | Easy-to-Hard | 35.2 (0.9) | 51.4 (0.5) |
| | FLYP + DisCL | Random | 35.9 (0.1) | 52.1 (0.2) |
| | FLYP + DisCL [Lower CLIPScore Threshold] | Adaptive | 37.1 (0.8) | 50.9 (0.9) |
| | FLYP + DisCL [Higher CLIPScore Threshold] | Adaptive | 38.1 (1.3) | 52.8 (0.8) |
| Ours | FLYP + DisCL | Adaptive | **38.2 (0.5)** | **54.3 (1.4)** |
| | FLYP + DisCL + WE | Adaptive | **38.7 (0.4)** | **54.6 (0.7)** |

iWildCam). For IN-LT results in Table 1, *All-Level Guidance* brings $\sim$1.27% gain in few-class accuracy, alongwith significant gains across other class-categories. Likewise, *All-Level Guidance* shows a superior ID and OOD performance as compared to *Text-only Guidance* for the iWildCam as well, see Table 5. These findings corroborate that utilizing a spectrum of data with multiple guidance levels helps mitigate the negative effects of the distribution gap.

## 6.2 EFFECT OF CURRICULUM LEARNING STRATEGY

**Long Tail Classification** We compare the impact of our *Diverse to Specific* curriculum strategy tailored for IN-LT task against other strategies, notably *All-Level Guidance* which employ no curriculum and uses all synthetic data. The *Diverse to Specific* demonstrate a higher few-class accuracy with a margin of 4.47%, see Fig. 3b. We then compare it with a reverse strategy *Specific-to-Diverse*, and found the latter one to be worse. The reverse strategy can overfit model to real distribution early on, increasing the gap between real and synthetic data; hence, later-stage training on the data with larger distribution gap can decrease models' few-class accuracy. For IN-LT, we also try *Adaptive* strategy (mainly developed for learning from low-quality data), in which strategy's progression is based on a validation set, comprising few tail images sampled from each guidance scale and few original images. But, validation set is scarce interms of tail samples, which renders it ineffective for identification of truly useful guidance. Hence, this strategy ranks as the least effective for LT task.

**Learning from Low Quality Data** For iWildCam task, we study the effect of our designed *Adaptive* strategy, catering to the challenge of learning from low quality data. As shown in Fig. 3d, for this task, *Adaptive* surpasses the *All-Level Guidance* with a clear margin, underscoring the benefit of using progressive curriculum over using all synthetic data. Further comparisons with the Non-Adaptive curricula including *Easy-to-Hard* and *Random*, show an impactful increase in OOD F1, while using our *Adaptive*.

These findings highlight how the structured data selection used in *Diverse-to-Specific*, is more effective in directing model's focus on scarce data (classes), however, when dealing with real-world low-quality data, an *Adaptive* strategy is more successful in adjusting to models' needs by adaptively selecting the suited data.

## 6.3 EFFECT OF CLIPSCORE THRESHOLD

**Long Tail Classification** Our analysis of CLIPScore distribution on IN-LT generated data leads us to infer that the best CLIPScore threshold for filtering is 0.3 (detailed explained in the Appendix A.1.2). We then assess different CLIPScore thresholds with the *Diverse to Specific* curriculum strategy, by experimenting with different values: lower (0.28), and higher (0.32), shown in Fig. 3a. However, we find that changing the CLIPScore threshold does not significantly affect the performance. As shown in Figure 4b, the CLIPScore of synthetic data is concentrated, as Stable Diffusion model performs

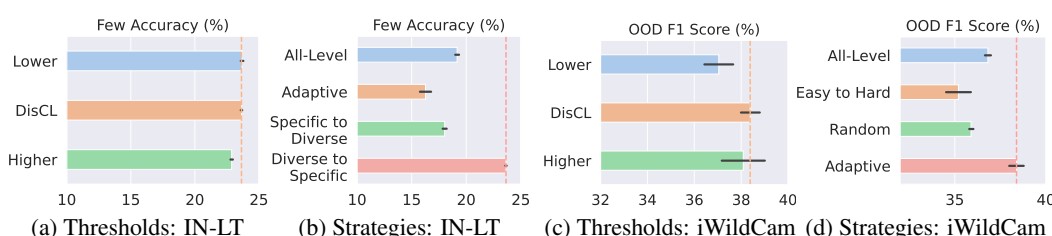

Figure 3: Ablation study of CLIPScore Thresholds (a,c) & Curriculum Strategies (b,d) on ImageNet-LT and iWildCam. The error bar reports the standard deviation of each experiment.

well on generating high-quality images for ImageNet classes. Changes in the CLIPScore threshold will not significantly affect the quality of synthetic images and corresponding effects in downstream classification tasks.

**Learning from Low Quality Data**  In the iWildCam task, we identify the optimal threshold as 0.25. To further validate this choice, we experiment with nearby thresholds (0.23 and 0.27) with the chosen *Adaptive Curriculum* strategy suited for low-quality image classification. As depicted in Fig. 3c, the 0.25 threshold markedly improves OOD performance compared to other CLIPScore thresholds. Unlike the ImageNet dataset, the iWildCam images are characterized by significant difficulty and poor quality, leading to high variance in CLIPScores of synthetic data (as shown in Fig. 5b). In this scenario, adjusting the CLIPScore threshold can impact model performance. When a higher threshold is used, the selected synthetic images include more prototypical visual features but they are less similar to the original images. Hence, they improve OOD performance but lead to a drop of ID F1 score.

The ablation study results on two classification tasks demonstrate that the selection of the CLIPScore threshold should be carefully aligned with the generation quality inherent to the task-at-hand.

## 7 CONCLUSION

In this paper, we introduce DisCL, a novel paradigm designed to enhance model performance when dealing with low-quality or scarce data. DisCL effectively bridges the distribution gap between original and target data using a spectrum of synthetic data, particularly for challenging samples. Our method utilizes image guidance in diffusion models to generate a comprehensive range of interpolated data from synthetic to real. Additionally, we design specific curricula to maximize the benefits of synthetic data for learning hard samples and closing the gap between synthetic and real data. The efficacy of DisCL is demonstrated through its significant and robust performance improvements in long-tail classification and learning from low-quality data, across various base model settings. Our analyses reveal that the interpolation of synthetic-to-real data, the selection of guidance intervals, and the proposed curriculum strategy are all essential components contributing to these gains.

Despite the promising results, the performance of DisCL is influenced by certain limitations. The quality of the generated data spectrum is dependent on the capabilities of the diffusion model and the visual-text alignment ability of filtering models. These dependencies constrain the overall performance of DisCL. Additionally, the current approach to generate text prompts for long-tail classification relies solely on category names derived from large language models (LLMs). To better align with the real data distribution and to reduce the gap between synthetic and real data, future works could focus on generating text prompts from image captions. Lastly, discrepancies in the position and size of class objects between real and synthetic images can widen the distribution gap. Addressing this issue may involve detecting objects and performing crop operations on real images or using detailed prompts to control these properties in synthetic data. These areas present opportunities for further research and improvement.

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

# A   APPENDIX / SUPPLEMENTAL MATERIAL

## A.1   SYNTHETIC DATA GENERATION WITH IMAGE GUIDANCE

In this section, we visualize more generated images in (Phase 1) of our method with various levels of image guidance, for two different classification tasks.

### A.1.1   GENERATION SETTINGS AND STATISTICS

We provide the statistics for the synthetic data generation within our paradigm on ImageNet-LT, CIFAR100-LT, iNaturalist2018, and iWildCam, as shown in Table 6.

Table 6: Statistics about Generated Synthetic Data. Irb refers to the imbalance ratio used to sample CIFAR100-LT dataset.

| Images' Details | ImageNet-LT | CIFAR100-LT Irb=100 | CIFAR100-LT Irb=50 | iNaturalist2018 | iWildCam |
|---|---|---|---|---|---|
| No. of Hard Samples | 1643 | 324 | 268 | 44956 | 8260 |
| Number of Image Guidance Scales $\lambda$ | 4 | 4 | 4 | 4 | 3 |
| Number of Random Seed Per Image | 8 | 8 | 8 | 4 | 8 |
| Number of Generated Images | 51917 | 2592 | 2144 | 179824 | 197756 |
| Number of Generated Images After Filtering | 24141 | 809 | 668 | 75234 | 90093 |

### A.1.2   IMAGENET-LT SYNTHETIC GENERATION

**Selection of Text prompts**   To improve model performance on the minority classes, high-quality and diverse synthetic samples are required. To achieve so, we follow the approach in Fu et al. (2024), and utilize publicly available GPT-3.5-turbo to generate diverse prompts for these 1000 IN-LT classes. We use the following prompt to query GPT-3.5-turbo for generating descriptions for class $X$:

"*Please provide 10 language descriptions for random scenes that contain only the class $X$ from the ImageNet-LT dataset. Each description should be different and contain a minimum of 15 words. These descriptions will serve as a guide for Stable Diffusion in generating images.*"

The sample-prompts generated by GPT-3.5-turbo are listed in Table 7.

**Selection of Images Guidance Levels**   We first analyze the cosine similarity between synthetic images and real images, as well as between synthetic images and text prompts. The similarity score between synthetic images and real images can be used to quantify the diversity introduced in the synthetic images. As depicted in Fig. 4a, the similarity between synthetic images and real images decrease as the guidance level reduces, demonstrating the trend of increased diversity in the data spectrum. However, the changes in the scores are relatively small across varying guidance levels. Combined with the visual cases for this dataset (examples shown in Fig. 6), we observe that for images generated with high guidance levels ($\lambda \geq 0.7$), only minor details are modified by the diffusion model,

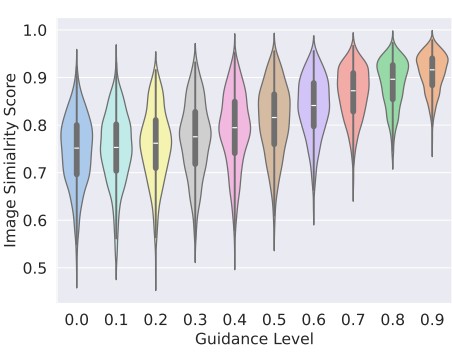
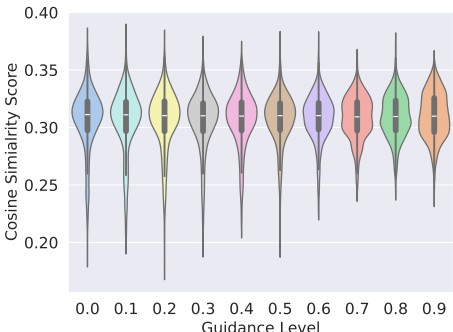

Figure 4: CLIP Cosine similarity score on ImageNet-LT computed between: (a) Synthetic image - original real images. (b) Synthetic image - defined text prompt.

Table 7: Generated text prompts for ImageNet-LT classes

| Class Name | Prompts |
|---|---|
| Grand Piano | A grand piano sits elegantly in a sunlit room, its glossy finish reflecting the warm glow. In a cozy living room, the grand piano adds a touch of luxury and sophistication to the space. The grand piano sits silently in a dimly lit room, waiting patiently for a skillful pianist to bring it to life. In a grand ballroom, the grand piano provides a majestic backdrop for a glamorous event. A vintage grand piano exudes timeless elegance in a quaint parlor, filled with antique charm. |
| Pufferfish | A colorful pufferfish swimming gracefully in a crystal-clear ocean, surrounded by vibrant coral reefs. A group of playful pufferfish blowing bubbles and chasing each other in a sunlit underwater cave. A shoal of pufferfish moving in unison, creating a mesmerizing dance of synchronized swimming in the deep sea. A fierce pufferfish defending its territory from intruders, puffing up its body and displaying its sharp spikes as a warning. A baby pufferfish following its larger parent closely, learning the ropes of survival in the vast ocean ecosystem. |

resulting in high similarity scores above 0.85. However, we aim to provide more diverse synthetic data to increase the model's generalization on the class-balanced test set. Including these highly similar images may hinder the diversity and cause the model to overfit to specific visual features, thereby negatively impacting its generalization ability. Therefore, we select $\{0.0, 0.1, 0.3, 0.5\}$ as the interval of image guidance levels used in the training process for this dataset.

**Selection of CLIPScore Threshold**  We leverage the widely used CLIPScore (Hessel et al., 2022) to filter out poor-quality images in the synthetic data. In this method, the CLIP cosine similarity between synthetic images' embeddings and text embeddings is computed to measure the alignment between images and the corresponding classes provided in text prompts. For the synthetic data generation for ImageNet-LT, we use a unified template that emphasizes the class information in text prompts. Following Trabucco et al. (2023), we use "*a photo of <class name>*" to prompt the CLIP model and compute the cosine similarity. We also consider the value of the filtering threshold for synthetic data. Following previous work (Schuhmann et al., 2021), we set the threshold to 0.3 based on the distribution of similarity scores and a review of generation quality, as shown in Fig. 4b. We observe that a threshold of 0.3 effectively filters out synthetic images with poor quality or mismatched classes.

### A.1.3 iWildCam Synthetic Generation

**Selection of Text prompts**  Following previous work (Clark & Jaini, 2024; Trabucco et al., 2023), we first define prompts for each class using the template "*a photo of <class>*". However, the classnames in iWildCam comprises of scientific names, which are usually unseen/unknown concepts to the diffusion text encoder. For example, "canis lupus" is the class name for "wolf" animal. To address this, we replace the scientific names with their common names and add a postfix "*in the wild*" in the prompt to drive the generation of wild images. The final text prompt we use is "*a photo of <common name of class> in the wild*".

**Selection of Images Guidance Levels**  Based on the generated data with multiple image guidance scales, we search for effective image guidance scales for this task using CLIP cosine similarity scores between synthetic image embeddings and real image embeddings. As shown in Fig. 5a, as the difference between real images and synthetic images increases, the cosine similarity between image embeddings decreases from $\lambda = 1$ to $\lambda = 0.3$. However, when the image guidance continues to decrease to $\lambda = 0$, the cosine similarity score increases slightly. With low image guidance scales, the diffusion model tends to generate images that heavily rely on text information, maintaining only global information (such as the color of the image background) in the synthetic data for some images.

This creates a distribution gap between these synthetic data and real data that is too large for the model to accurately compare the differences between the two images using embedding representation. Additionally, based on the analysis of the quality of synthetic images and to leverage the difficulty of the features and the distribution gap between synthetic and real data, we set the image guidance scales to $\{0.5, 0.7, 0.9\}$ for this task.

**Selection of CLIPScore Threshold** To filter out low-quality images, we assess the CLIP cosine similarity scores between synthetic image embeddings and corresponding text embeddings for each class. We use the same prompt template as in the generation process ("*a photo of <common name for animal> in the wild*") to compute CLIPScore for synthetic images. The distribution of CLIPScores is shown in Fig. 5b, which reveals a distinct gap around 0.25. Combined with a review of the quality of synthetic data, we set the threshold to 0.25. Synthetic data with a CLIPScore lower than 0.25 are considered poor-quality samples.

### A.1.4 VISUALIZATION

**Visual Cases** We provide additional visual examples of synthetic data generated with multiple guidance levels and text prompts for the ImageNet-LT and iWildCam datasets. The results are visualized in Fig. 6 and Fig. 7. These examples demonstrate that the model can generate synthetic data with various postures, backgrounds, and actions as the image guidance level decreases. Particularly for ImageNet-LT generation results, diverse prompts introduce more varied features into low-guidance data. These diverse features enable the model to achieve better generalization on the target distribution.

**Failure Cases** During generation, despite designing text prompts and applying CLIPScore to filter to remove low-quality data, some failure cases still occur in the synthetic dataset. In this section, we discuss these failure cases encountered during the generation process. As shown in Fig. 8 and Fig. 9, the first failure case is caused due to the inability to recognize objects in the original images. If these objects are clearly obscured or hard-to-identify (e.g. second case in Fig. 9 and first case in Fig. 8), diffusion models cannot accurately identify the object or modify details for generating diverse and useful data. For these seed images, only synthetic data generated with a low-guidance scale can achieve a CLIPScore higher than the threshold. However, this approach compromises the smooth transition of data from synthetic to real distribution. Even though the diffusion model can generate images with a smooth transition for most-of-the-cases, our quality-check on synthetic data can constrain the feature extraction and alignment ability of the CLIP model. For example, in second case of Fig. 8, CLIPScore filters out the slightly modified but perceptually useful images, containing prototypical class features.

### A.2 APPLICATIONS ON OTHER DATASETS

To further evaluate the robustness of DisCL, we extended our experiments to two additional widely used imbalanced datasets: CIFAR-100-LT (Cao et al., 2019) and iNaturalist2018 (Van Horn et al., 2018). For iNaturalist2018, We generated synthetic data for these datasets following the same approach and settings used for the long-tail classification task on ImageNet-LT. For CIFAR-100-LT

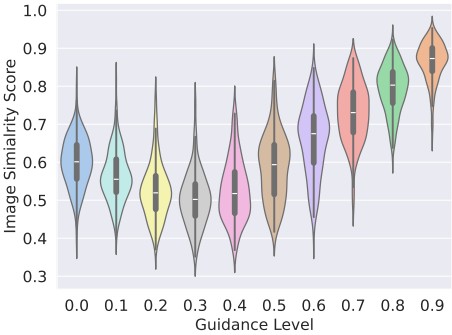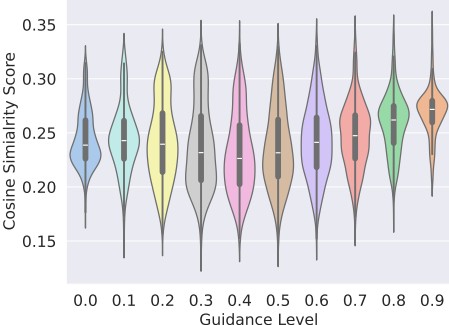

Figure 5: CLIP Cosine similarity score for iWildCam Synthesis. (a) Synthetic image & original real images. (b) Synthetic image & defined text prompt.

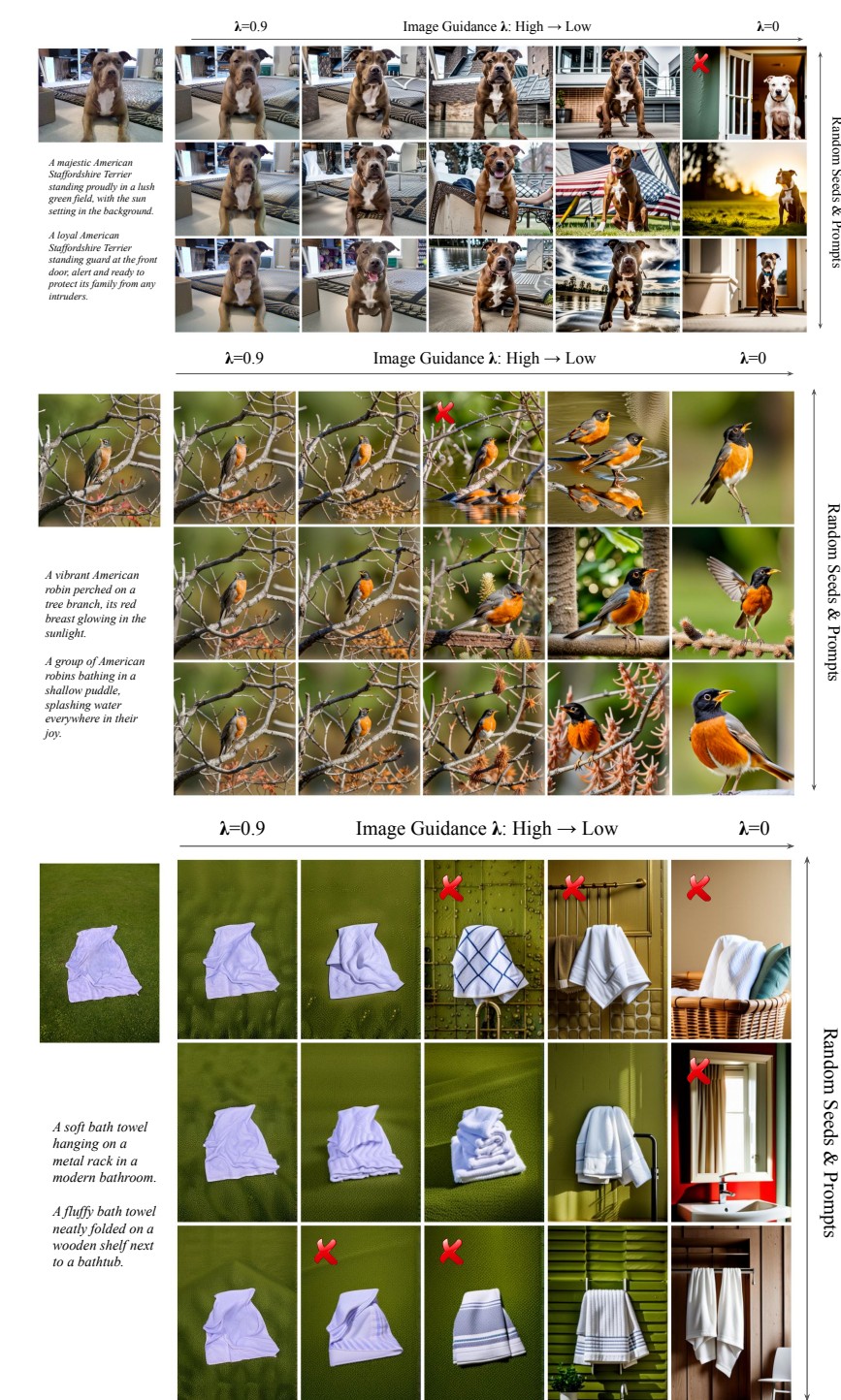

Figure 6: Synthetic generation with various image guidance and random seeds based on ImageNet-LT.

dataset, due to the low resolution of the original images, we adjust the image guidance scale to $0.5, 0.7, 0.9$ to ensure generation quality for the synthetic data. Visual examples of the generated data are shown in Fig. 10 and Fig. 11. For CIFAR-100-LT, we assessed the performance of DisCL across different imbalance ratios (50 and 100). The results, along with those of the baseline methods, are presented in Table 2 and Table 3. Our experimental findings demonstrate that DisCL achieved significant improvements in Top-1 accuracy for both overall and few-shot classes across these datasets.

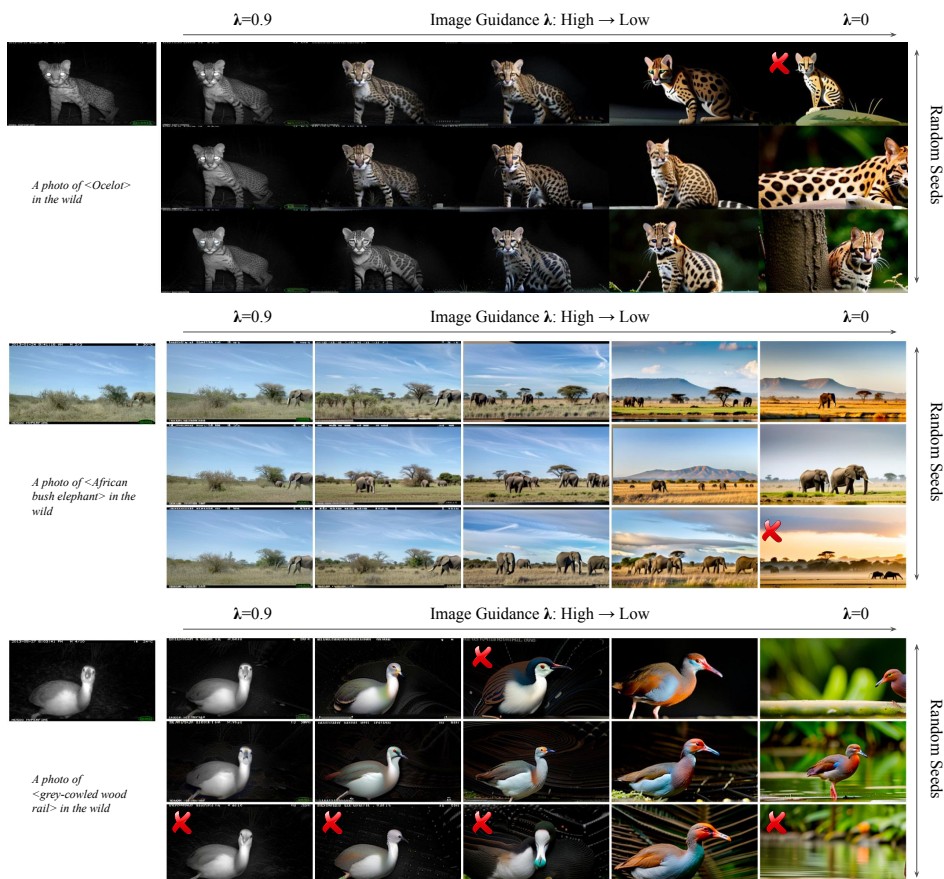

Figure 7: Synthetic generation with various image guidance and random seeds based on iWildCam.

## A.3 TRAINING WITH CURRICULUM LEARNING

### A.3.1 LONG-TAIL CLASSIFICATION WITH NON-ADAPTIVE STRATEGY

For long-tail classification, we propose a non-adaptive curriculum learning strategy that starts with the lowest guidance and progressively increases to the highest guidance within the defined interval $\Lambda$. We employ a linear scheduler to adjust the guidance levels during training, allowing the model to train with data from various guidance levels for *equal durations*. Furthermore, the test set of ImageNet-LT is in-distribution to its training data; unlike the training data, it is a class-balanced set. To mitigate the potential negative effects of the distribution gap between synthetic and real data, all the hard tail samples from original data are involved into training at all times. Furthermore, with DisCL, number of samples for tail classes increases along with the introduction of synthetic data at each stage, however the ratio of tail-to-nontail samples is still very skewed. To preserve a constant imbalance-ratio throughout all training stages and experiments, we undersample the non-tail samples at "each stage" so that ratio of tail-samples to non-tail samples matches the proportion of tail classes to non-tail classes present in the original data (13.6%).

All experiments are conducted based on this proportion setting. Complete strategy details are covered in Algorithm 1.

### A.3.2 LEARNING FROM LOW-QUALITY DATA WITH "ADAPTIVE CURRICULUM" STRATEGY

An approximation method to assess the effectiveness of samples in helping model achieve greatest progress on and fastest learning face is introduced by DoCL (Zhou et al., 2021a) as shown in Eq 4.

$$\mathbb{E}_{x \in D, x \sim \mathcal{D}} \langle y - f(x), \frac{\partial f(x)}{\partial t}|_S \rangle \approx \frac{1}{|D|} \sum_{i \in \mathcal{V}} \langle y_i - f(x_i), \frac{\partial f(x_i)}{\partial t}|_D \rangle \tag{4}$$

where $\mathcal{D}$ is the training distribution and $x \in D$ is a set of finite samples randomly sampled from the original distribution $\mathcal{D}$. $\mathcal{V}$ denotes the subset of samples. Here, $y$ and $f(x)$ denotes the target-class

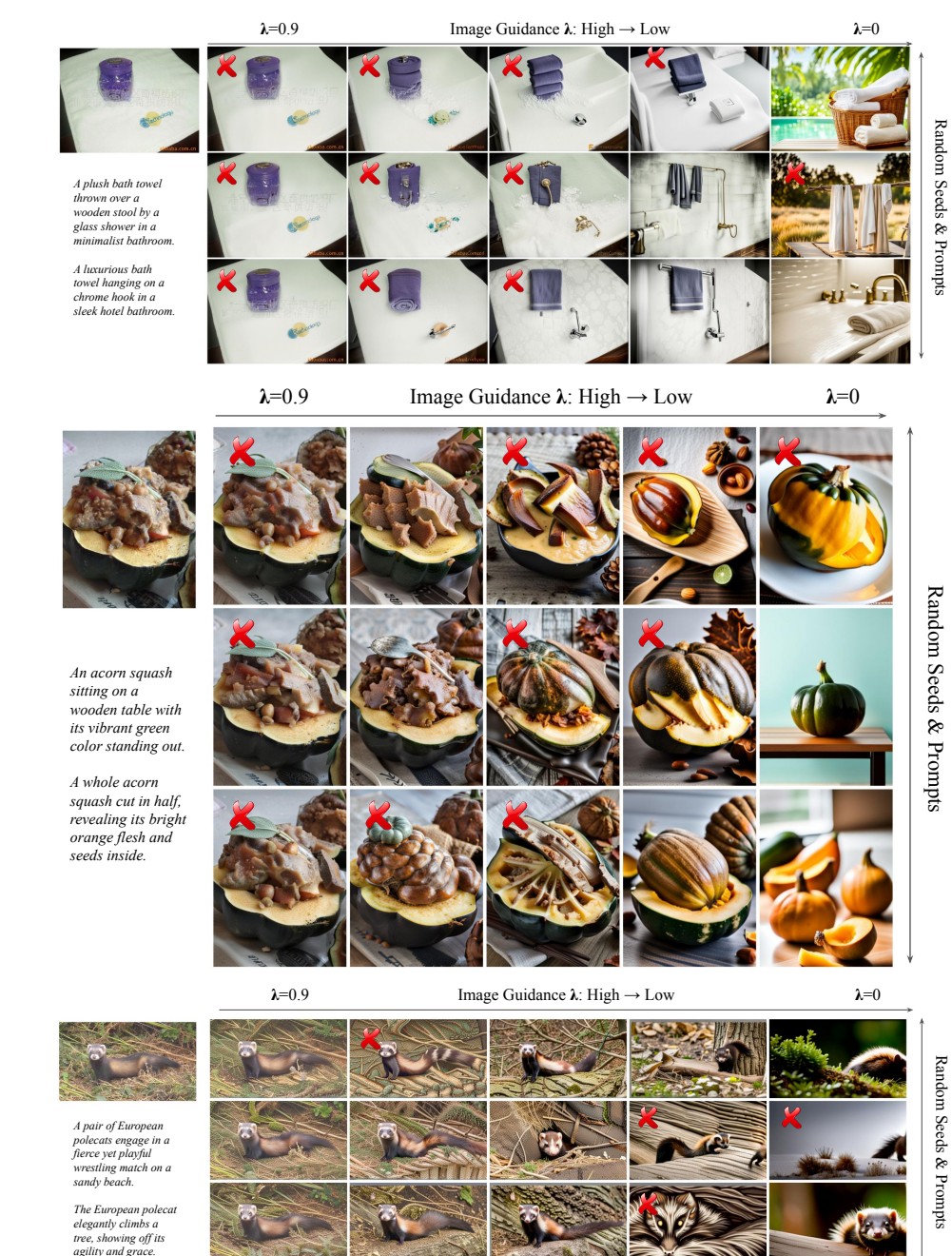

Figure 8: Failure cases for ImageNet-LT synthetic generation

and sample prediction. $\langle y - f(x), \frac{\partial f(x)}{\partial t} |_{\mathcal{V}} \rangle$ represents the project of residual $y - f(x)$ on the model dynamics $\frac{\partial f(x)}{\partial t}|_{\mathcal{V}}$. This equation indicates that when trained with subset $\mathcal{V}$, the expected progress $\mathbb{E}$ of samples in the original training dataset can be approximated by the progress of samples on subset $\mathcal{V}$ achieved via training on the set $D$.

For learning from low-quality data, we adopt DoCL and implement an adaptive curriculum strategy to select the synthetic data with best guidance level for each training stage. Before the training process, we randomly select samples from the spectrum for each guidance level in $\Lambda$ and mark it as guidance validation set $\mathcal{V}$ for progress evaluation. This set has zero overlap with the training data $\mathcal{D}_{\text{all}}$. At each training stage, we randomly sample a set $D$ (termed as random-real set) from the training dataset $\mathcal{D}_{\text{all}}$. Before selecting the guidance level, we train the model on dataset $D$ and evaluate the progress (in

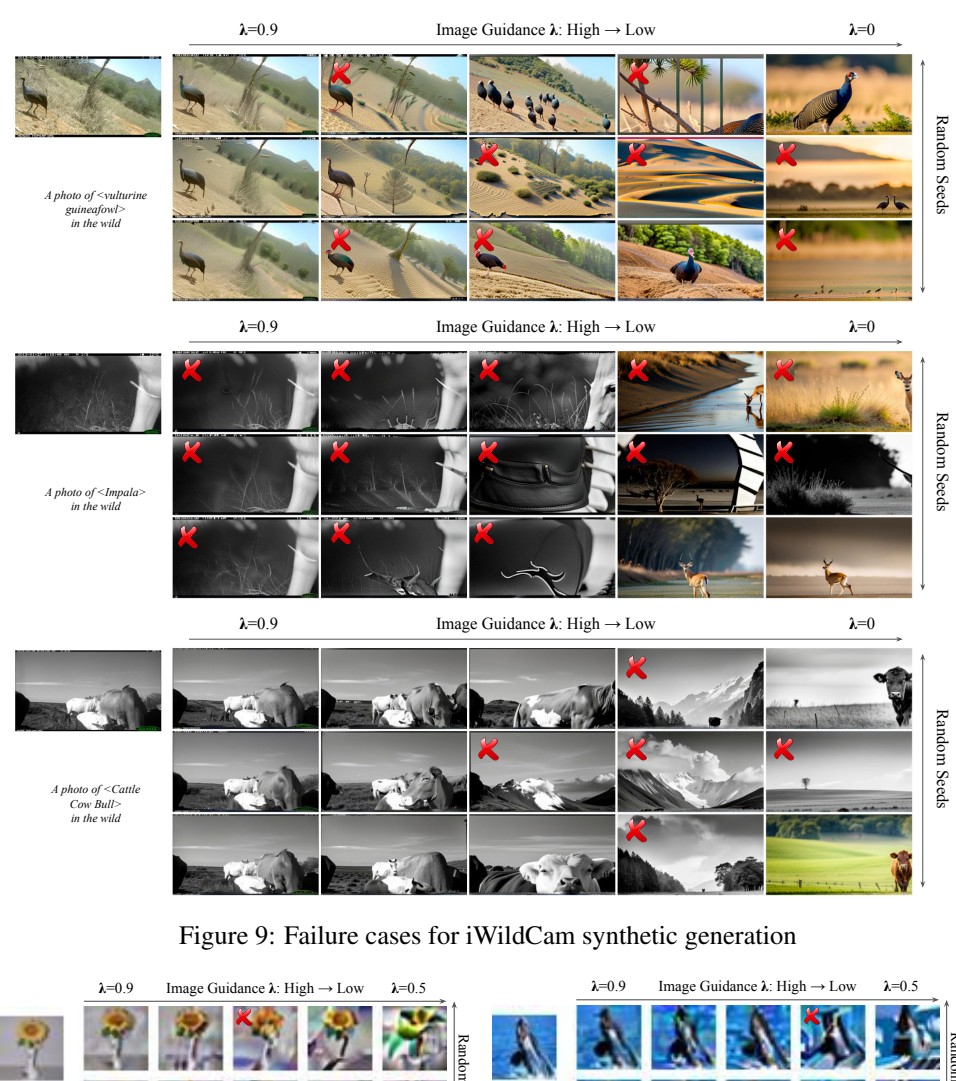

Figure 9: Failure cases for iWildCam synthetic generation

Figure 10: Synthetic generation with various image guidance and random seeds based on CIFAR100. Sample Prompt: *(1) A bright sunflower standing tall in a field, basking in the warm sunlight of a summer day. (2) A majestic whale breaches the surface of the deep blue ocean, sending a spray of water into the air.*

terms of classifier's prediction score) achieved on samples of each subset $\mathcal{V}_i$ corresponding to a given guidance $\lambda_i$. We then select the $\lambda_i$ with the highest progress to gather synthetic data and combine it with other non-hard samples from the original training data for the current training stage. This technique encourages the model to adaptively select the most informative guidance for the current training stage. At the end of the curriculum-training, to alleviate the negative effect of the distribution gap between synthetic data and real data for this task, we keep finetuning the model with real data for a short period. The steps of algorithm are detailed in Algorithm 2.

## A.4 HYPERPARAMETERS FOR SYNTHETIC GENERATION AND MODEL TRAINING

The values of all hyperparameters used for synthetic data generation with diffusion model and curriculum learning strategy are listed in Table 8.

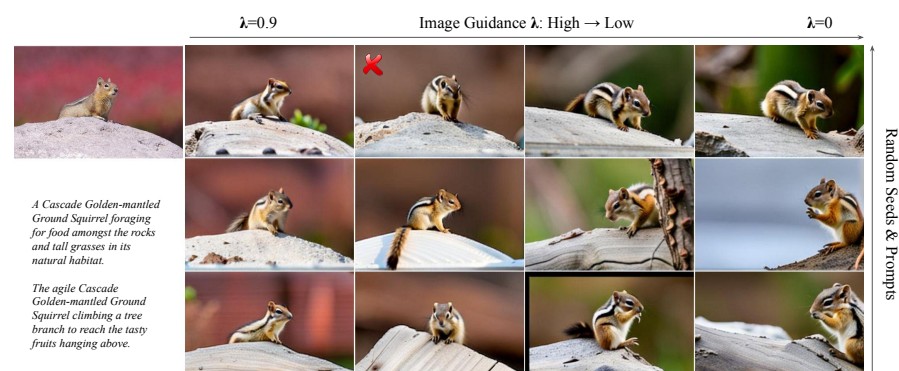

Figure 11: Synthetic generation with various image guidance and random seeds based on iNaturalist 2018.

For ImageNet-LT, we implement baselines based on the codebase and the pretrained model from LDMLR. We also re-implement CUDA baseline from this codebase, containing some missing models. We use the same hyper-parameter settings as listed in the CUDA paper. For FLYP, we implement baseline models with FLYP codebase and leverage the available pretrained model from Open CLIP.

### A.5 COMPUTATIONAL REQUIREMENTS FOR SYNTHETIC GENERATION

For computational requirements of offline generation, 1 RTX A5000 GPU is used to generate synthetic images. For time efficiency, It took 10 seconds to generate a full spectrum (6 image guidance levels) of synthetic images for each real image with resolution=$480 \times 270$.

### A.6 FURTHER DISCUSSION ON EXPERIMENT RESULTS

In this section, we analyze the results of each guidance level under *Fixed Guidance* experiment to observe the effect of different image guidance levels on the classifier's performance. During the training process, synthetic data generated from only a specific guidance level combined with original real data is presented to the model. The ablation numbers are shown in Fig. 12.

For the iWildCam dataset, data generated with text-only guidance ($\lambda = 0$) has the largest distribution gap between synthetic and real data, and it also showcases lowest Out-of-Distribution (OOD) performance. As the guidance scale increases, this distribution gap diminishes, and the OOD F1 score consistently improves. This outcome aligns with the visually observed reduction in distribution differences between generated and real images.

---

**Algorithm 1:** Training with **non-adaptive** Curriculum strategy

---

**Input:** Image guidance level $\Lambda = \{\lambda_i | \lambda_i \in [0, 1]\}$, non-hard samples $\mathcal{D}_{nh} = \{(x_i, y_i, \lambda_i = 1)\}_{i=1}^N$,
spectrum of syn-to-real data $\mathcal{S} = \{(x'_j, y_j, \lambda_j) | \lambda_j \in \Lambda\}_{j=1}^M$, original hard samples
$\mathcal{D}_h = \{(x_j, y_j, \lambda_j) | \lambda_j = 1, (x_j, y_j, \lambda_j) \in \mathcal{S}\}$, train epochs $E$, curriculum epochs $E_{CL}$,
predefined Linear Guidance Schedule $\mathcal{G} = \{\lambda_1, \lambda_2, ..., \lambda_e, ..., \lambda_{E_{CL}}\}$.
**Output:** trained model $f_\theta$
**Initialize:** pretrained model $f_\theta$

**1 for** $e \leq E_{CL}$ **do**
**2**     $\lambda_e = \mathcal{G}(e)$
**3**     Extract $\mathcal{S}_{\lambda_e} = \{(x_j, y_j, \lambda_j) | \lambda_j = \lambda_e\}$
**4**     Gather new training set $\mathcal{D}_e = \mathcal{S}_{\lambda_e} \cup \mathcal{D}_{nh} \cup \mathcal{D}_h$
**5**     Finetune the model $f_\theta$ with $\mathcal{D}_e$
**6 end**
**7 for** $e > E_{CL}$ *and* $e \leq E$ **do**
**8**     Gather new training set $\mathcal{D}_e = \mathcal{D}_{nh} \cup \mathcal{D}_h$
**9**     Finetune the model $f_\theta$ with $\mathcal{D}_e$
**10 end**

---

---

**Algorithm 2:** Training with **adaptive** Curriculum strategy

---

**Input:** Image guidance level $\Lambda = \{\lambda_i | \lambda_i \in [0,1]\}$, non-hard samples $\mathcal{D}_{\text{nh}} = \{(x_i, y_i, \lambda_i = 1)\}_{i=1}^N$,
  spectrum of syn-to-real data $\mathcal{S} = \{(x'_j, y_j, \lambda_j) | \lambda_j \in \Lambda\}_{j=1}^M$, original training data
  $\mathcal{D}_{\text{all}} = \mathcal{D}_{\text{nh}} \cup \{(x'_j, y_j, \lambda_j) | \lambda_j = 1\}$, guidance validation set $\mathcal{V} = \{(x'_j, y_j, \lambda_j) | \lambda_j \in \Lambda\}_{j=1}^m$, train
  epochs $E$, curriculum epoch $E_{CL}$, size of random-real set $|D|$.
**Output:** trained model $f_\theta$
**Initialize:** pretrained model $f_\theta$
/* Note: Set $\mathcal{V}$ has no overlap with $\mathcal{D}_{\text{all}}$.          */

1   **for** $e \leq E_{CL}$ **do**
2      Calculate true-class probability $p_{\text{bef}}$ of model $f_\theta$ on set $\mathcal{V}$
3      Sample a random-real set $D$ from $\mathcal{D}_{\text{all}}$ /* contains Real data only       */
4      Training model $f_\theta$ with $D$
5      Calculate true-class probability $p_{\text{aft}}$ of model $f_\theta$ on set $\mathcal{V}$
6      $\lambda_e \leftarrow \arg\max_{\lambda_i \in \Lambda} (p_{\text{aft}} - p_{\text{bef}})$
7      Extract $\mathcal{S}_{\lambda_e} = \{(x_j, y_j, \lambda_j) | \lambda_j = \lambda_e\}$
8      Gather new training set $\mathcal{D}_e = \mathcal{S}_{\lambda_e} \cup \mathcal{D}_{\text{nh}}$
9      Train the model $f_\theta$ with $\mathcal{D}_e$
10 **end**
11 **for** $e > E_{CL}$ *and* $e \leq E$ **do**
12      Train the model $f_\theta$ with $\mathcal{D}_{\text{all}}$
13 **end**

---

Conversely, the trend seen with ImageNet-LT diverges from above. In long-tail classification, we aim to increase data diversity while keeping the distribution gap small. As detailed in Appendix A.1.2, on one hand, generating synthetic data that closely resemble real data further reduces the diversity, and generating synthetic data far from real distribution can offer diversity but hurt OOD performance. In case of ImageNet-LT, we observe that more diverse synthetic data tends to significantly improve the classifiers' generalization.

Inspired by these observations, we tailor our guidance scales intervals according to the task-at-hand.

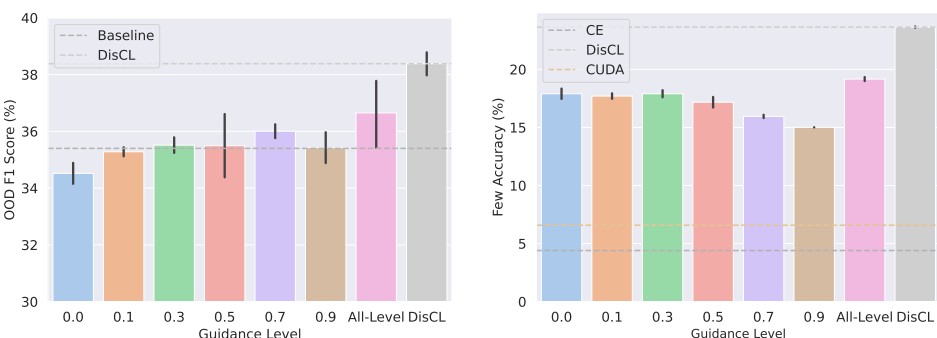

Figure 12: Effect of Image Guidance (mixing syn+real). All-level experiments use the synthesis samples from all guidance scales selected for each task. $0.5$ refers to only using synthetic data with guidance level $\lambda = 0.5$ for fine-tuning. Left: results on iWildCam. Right: results on ImageNet-LT

A.7   SOCIETAL IMPACT

Our proposed method is beneficial for diverse fields, where inadequate quantity and low quality of data is common, *e.g.* medical domain. The synthetic data generation, as followed by DisCL approach can reduce the need for extensive data collection, therefore mitigating the ethical concerns related to data-privacy. Overall, our method DisCL can democratize the access of effectively training ML models in the low-resource environments. However, by leveraging the pretrained generative models, the potential biases of models can perpetuate into the synthetic data and eventually affect the sensitive real-world applications consuming this data, such as medical diagnosis, law enforcement *etc.*

Table 8: Hyperparameters and their values

| | Hyperparameter Name | Value |
|---|---|---|
| **Generation** | Text Guidance Scale $w$ | 10 |
| | Noise Scheduler | DDIM |
| | Stable Diffusion Denoising Steps | 1000 |
| | Stable Diffusion Checkpoint | stabilityai/stable-diffusion-xl-refiner-1.0 |
| | CLIP Filter Model | openai/clip-vit-base-patch32 |
| | Filtering Threshold for iWildCam | 0.25 |
| | Filtering Threshold for ImageNet-LT | 0.30 |
| | GPU Used | Nvidia rtx5000 with 24GB |
| **ImageNet-LT** | Level of Image Guidances $\lambda$ | $\{0, 0.1, 0.3, 0.5, 1.0\}$ |
| | CLIP Filtering Threshold | 0.3 |
| | Batch Size for ResNet-10 | 128 |
| | Learning Rate | $1e$-3 |
| | Optimizer | Adam |
| | Scheduler | Cosine |
| | Training Epoch | 65 |
| | Training Epoch for Curriculum Learning | 60 |
| | GPU Used | Nvidia rtx5000 with 24GB |
| **iWildCam** | Level of Image Guidances $\lambda$ | $\{0.5, 0.7, 0.9, 1.0\}$ |
| | CLIP Filtering Threshold | 0.25 |
| | Size of Dataset $D$ | 30000 |
| | Size of Guidance Validate Dataset $S$ | 2000 |
| | Batch Size for CLIP ViT-B/16 | 256 |
| | Batch Size for CLIP ViT-L/16 | 200 |
| | Learning Rate | $1e$-5 |
| | Optimizer | AdamW |
| | Scheduler | Cosine with Warmup |
| | Warmup Step | 500 |
| | Training Epoch | 20 |
| | Training Epoch for Curriculum Learning | 15 |
| | GPU Used | 2 Nvidia A100 with 80GB |

