# OpenReview forum: "Diffusion Curriculum: Synthetic-to-Real Data Curriculum via Image-Guided Diffusion"
_ICLR.cc/2025/Conference — ICLR 2025 Conference Withdrawn Submission_

### Official Review · Reviewer_9SrJ · 2024-10-28

**Soundness:** 3
**Presentation:** 3
**Contribution:** 3
**Rating:** 5
**Confidence:** 4

**Summary:**

The paper addresses the challenge from training computer vision models (image classification task) with low-quality or scarce data. The paper proposes Diffusion Curriculum (DisCL) which leverages diffusion models to synthesize hard image examples data with different guidance scales and then utilizes a Generative Curriculum Learning to select appropriate synthetic data from the full spectrum of generated data for training data augmentation.  Experiments are conducted on two tasks: long-tail classification and learning from low-quality data, to show the method's effectiveness.

**Strengths:**

1. The idea of adjusting guidance scales to obtain a greater variety and quality of training data augmentation is interesting and novel.
2. Generative Curriculum Learning is reasonable and can adjust for different tasks.
2. The proposed method proves its effectiveness in both experiments in long-tail classification and learning from low-quality data.

**Weaknesses:**

1. A tradeoff is the speed of generating a full spectrum for images.  For large datasets, the diffusion models can consume a long time to generate a full spectrum of needed images.

**Questions:**

I am concerned about using only the pre-trained stable diffusion model without further fine-tuning (I am happy to update the rating if the following questions are addressed).
1. Were there any observed biases/failures in the synthetic data generated for small resolution datasets (For example CIFAR100-LT)? I am concerned because of the resolution differences between the CIFAR100-LT dataset and the resolution stable diffusion model is trained on.
2. In real-life scenarios, some datasets we want to train the models are not real photographs, do you think a pre-trained diffusion model and your proposed method can be effective for Long-Tailed or low-quality datasets in domains of Comics, Drawings, etc?
3. Using Clip score as a threshold to filter out generated images is reasonable but for some classes, it can be easy to filter out too many generated data from pretrained stable diffusion (therefore not a full spectrum of data can be generated and saved). May I ask if this situation also occurs in your experiments and could you provide the percentage of filtered images and any strategies you employed to ensure sufficient data?

---

> ### Author Response · Authors · 2024-11-15
>
> Thank you for your valuable feedback! Here we address you comments.
>
> > Tradeoff between time and synthetic dataset.
>
> DisCL generates synthetic data exclusively for hard samples in the original datasets, covering approximately only 10% of total samples. Additionally, as noted in Appendix A.5, generating a complete spectrum of data across six image guidance levels (at a resolution of 480×270) takes only 10 seconds. This generation time is efficient and manageable within these parameters.
>
> > Failure cases for low-resolution datasets.
>
> For low-resolution datasets, image editing with diffusion models can be challenging. To address this, we start by using a super-resolved version of CIFAR100 images (resized with CAI super resolution) as the base for synthetic data generation. The generated images are then downsampled to 32×32 for training.
> Additionally, we observed that lower image guidance levels can produce significantly altered synthetic images, creating a large discrepancy between the synthetic and original images. To improve the quality of synthetic data for low-resolution datasets, we adjust the image guidance level to a range of 0.5 to 0.9, ensuring closer alignment with the original images.
>
> - 128*128 resized CIFAR100 dataset: https://www.kaggle.com/datasets/joaopauloschuler/cifar100-128x128-resized-via-cai-super-resolution
>
> > Application to non-realistic domains.
>
> For datasets in non-realistic domains such as Comics or Drawings, we can first evaluate the generation quality of the pre-trained diffusion model on the target dataset. If the quality is insufficient, we can switch to pre-trained models specifically trained on the corresponding or similar domains.
>
> In the future, we plan to integrate advanced diffusion training techniques, such as ControlNet and DreamBooth, to enhance the compatibility of diffusion models with the original dataset, ensuring higher-quality generation aligned with the domain’s characteristics.
>
> > Questions related to CLIPScore filtering process.
>
> We assess the effect of CLIPScore filtering with human evaluation. To ensure the quality of synthetic data, we choose a relatively high threshold of CLIPScore for data filtering, which results in a high ratio of abandoned data.
> For percentage of filtered images is shown as below, which is shown in Table 6 in Appendix.
>
> |  | Generated Image | Generated Image After Filteration | Acceptance Rate
> | --- | --- | ---| ---|
> | ImageNet-LT | 51917 |24141 | 46.50% |
> | CIFAR100-LT (irb=50) | 2592  | 809 | 31.21% |
> | CIFAR100-LT (irb=100) | 2144 | 668 | 31.16% |
> | iNaturalist2018 | 179824 | 75234 | 41.84% |
> | iWildCam | 197756 | 90093 | 45.56% |

---

### Official Review · Reviewer_VQ25 · 2024-10-28

**Soundness:** 2
**Presentation:** 2
**Contribution:** 2
**Rating:** 3
**Confidence:** 4

**Summary:**

Training deep learning models with low-quality or limited amounts of data often results in overfitting or suboptimal performance. To overcome this challenge, data augmentations have been an integral part of training deep learning models. However, classical data augmentations offer limited diversity and may also result in out-of-distribution samples, hampering the performance of the model. Therefore, recent research has focused on using generative models for data augmentations. Building in this direction, the authors propose a method to create a spectrum of interpolations between synthetic and real images called Diffusion Curriculum (DisCL). Focusing on the long-tail classification and learning from low-quality data tasks, the author demonstrates the efficacy of DisCL.

**Strengths:**

- The authors address the gap between real and synthetic data generated using diffusion models by designing a generative curriculum that can adjust the quality, diversity, and difficulty of the data for different training stages. This provides a new perspective on generative data augmentation, given that the majority of prior work considered a fixed image guidance scale throughout the training.
- The effectiveness of generative data augmentation strategies primarily depends on the performance of the generative model. To address the potential shortcomings of the generative models and thereby improve the effectiveness of the data augmentation, the authors proposed using CLIPScorer to filter out low-fidelity images.

**Weaknesses:**

- The paper is difficult to follow. This is partly because key details are missing in the main paper. For instance, in Section 4.1, the authors mention the use of a set of diverse textual prompts, while the details are deferred to the appendix. Another instance is in Section 4.2, where the authors mention that inspired by DoCL, they propose an adaptive curriculum. However, there is no discussion of the proposed adaptive curriculum in that section.
- The concept that the choice of the starting timestep $t$ controls the impact of $z_{real}$ has been extensively studied in prior works, notable being SDEdit [1]. Therefore, it would be easier for the readers to follow if the authors cite the existing works and explain the similarities.
- Using generative models for data augmentation has been an active area of research, with many approaches proposed in the literature [2,3,4,5]. The authors can compare their approach with these existing works to substantiate their novelty and demonstrate the impact of using a pre-defined or adaptive generative curriculum. The current evaluation is limited.

References:
- Meng, Chenlin, et al. "Sdedit: Guided image synthesis and editing with stochastic differential equations." arXiv preprint arXiv:2108.01073 (2021).
- Roy, Aniket, et al. "Cap2aug: Caption guided image to image data augmentation." arXiv preprint arXiv:2212.05404 (2022).
- Luzi, Lorenzo, et al. "Boomerang: Local sampling on image manifolds using diffusion models." arXiv preprint arXiv:2210.12100 (2022).
- Koohpayegani, Soroush Abbasi, et al. "GeNIe: Generative Hard Negative Images Through Diffusion." arXiv preprint arXiv:2312.02548 (2023).
- Trabucco, Brandon, et al. "Effective data augmentation with diffusion models." arXiv preprint arXiv:2302.07944 (2023).

**Questions:**

- Are ‘diverse to specific’ and ‘easy to hard’ curriculum strategies the same? If so, why are they called differently?
- In Table 1, for the “Few” class, the impact of DisCL is more significant when using Cross Entropy compared to when Balanced Softmax is used. Why?
- One of the benchmarks for learning from low-quality data is ALIA. Which Table contains the results with ALIA?

---

> ### Author Response · Authors · 2024-11-15
>
> Thank you for your feedback and suggestions! We provide responses to your questions below.
>
> > Detailed information for Section 4.
>
> Thanks for your advice! The details of DoCL and its adaptive curriculum are discussed in Appendix A.3.2. We will try our best to move back more details from the Appendix to the main paper.
>
> > Related works about using generative models for data augmentation.
>
> Thank you for pointing out these related works! We would like to clarify key distinctions between our approach and these methods.
>
> Our DisCL framework is built on two primary components: the generation of a **spectrum of sync-to-real synthetic data** and a **curriculum learning paradigm** designed to effectively leverage this data. Unlike previous works, which focus primarily on synthetic data generation for downstream tasks without integrating curriculum learning, DisCL strategically bridges the distributional gap between synthetic and real data.
> For example, Cap2Aug (Roy et al., 2022) generates data with fine-grained modifications, while Boomerang (Luzi et al., 2022) synthesizes data through local sampling to closely resemble the original inputs. Both methods focus on localized changes and thus lack the diversity needed for long-tail tasks and easier or typical samples needed for low-quality tasks.
> GeNIe (Koohpayegani et al., 2023) generates hard negative samples to improve the classifier’s ability to distinguish between positive and negative samples. However, for low-quality tasks like iWildCam, the primary challenge is aligning visual features of hard positive samples with pre-trained knowledge—an issue that GeNIe does not fully address. DA-Fusion (Trabucco et al., 2023) increases synthetic data diversity, yet it lacks either the spectrum of varied synthesis or the curriculum-driven approach that DisCL provides.
> DisCL’s curriculum-based framework constructs a smooth transition from synthetic data (representing prototypical or diverse features) to real data (which contains task-specific but often limited features). This smooth spectrum aids in adapting pre-trained models more effectively to low-quality or long-tail tasks.
> We plan to explore integrating image editing approaches from these works with DisCL’s spectrum generation and curriculum paradigm to further enhance our results.
>
> > Question related to ALIA results.
>
> The results of ALIA are shown below. We will update the result in Table 5.
>
> |  | OOD F1 Score | ID F1 Score |
> | --- | --- | --- |
> | ALIA | 36.9 (0.3) | 52.6 (0.4) |
> | DisCL | **38.2 (0.5)**  | **54.3 (1.4)**  |
>
> > Question related to curriculum strategies.
>
> The 'diverse to specific' and 'easy to hard' curriculum strategies both refer to 'synthetic (lower image guidance) to real (higher image guidance)' curriculum strategies (as mentioned in line 360 and 407). However, for different tasks, synthetic data has different properties. For long-tail task, synthetic data with lower image guidance provides more diversified visual features, while providing easier and more proto-typical features for low-quality tasks. To illustrate the property of synthetic data under different tasks, we use these two names.
>
>
> > Questions related to Cross Entropy and Balanced Softmax.
>
> Due to the population bias among classes, the model struggles to sufficiently learn visual features from underrepresented “Few” classes while using Cross Entropy loss function. As a result, the Cross-Entropy (CE) baseline performs significantly worse compared to the Balanced Softmax (BS) loss, which explicitly addresses class imbalance by reweighting softmax probabilities based on class frequency.
>
> When applying DisCL, our method introduces more diverse features for underrepresented classes, partially mitigating the effects of population bias and improving performance under CE. However, for Balanced Softmax, which already compensates for class imbalance directly, the additional impact of DisCL’s diverse features and samples is less pronounced.
>
> - Meng, Chenlin, et al. "Sdedit: Guided image synthesis and editing with stochastic differential equations." arXiv preprint arXiv:2108.01073 (2021).
> - Roy, Aniket, et al. "Cap2aug: Caption guided image to image data augmentation." arXiv preprint arXiv:2212.05404 (2022).
> - Luzi, Lorenzo, et al. "Boomerang: Local sampling on image manifolds using diffusion models." arXiv preprint arXiv:2210.12100 (2022).
> - Koohpayegani, Soroush Abbasi, et al. "GeNIe: Generative Hard Negative Images Through Diffusion." arXiv preprint arXiv:2312.02548 (2023).
> - Trabucco, Brandon, et al. "Effective data augmentation with diffusion models." arXiv preprint arXiv:2302.07944 (2023).

---

### Official Review · Reviewer_HFGS · 2024-10-30

**Soundness:** 2
**Presentation:** 2
**Contribution:** 2
**Rating:** 3
**Confidence:** 4

**Summary:**

This paper tries to incorporate the curriculum learning technique into image data augmentation.

This paper evaluated the proposed method on two tasks: long-tail classification and image classification with low-quality data to show the effectiveness.

Contribution: This paper reveals that curriculum learning is a way to balance synthetic data with various quality and real data.

**Strengths:**

1. The experiment is well designed.
2. The visualization is good.
3. A substantial improvement has been achieved for some tasks.
4. Combine the curriculum learning into generative data augmentation.

**Weaknesses:**

1. In line 87, "We harness image guidance in diffusion models to create a spectrum of synthetic-to-real data". I don't think this is a contribution of yours. In ICLR 2023, one paper called "IS SYNTHETIC DATA FROM GENERATIVE MODELS READY FOR IMAGE RECOGNITION?" has already proposed to leverage both image and text guidance for data augmentation, and there are a lot of following works.

2. In the method part, most words are recalling the diffusion theory and image-text guidance, which are both not your contribution. I think the main contribution is how to leverage the various quality data with curriculum learning. However, the Sec. 3.2 is quite short and simple.

3. In the ablation part of Table 1, compared with CE + Text-only Guidance (39.10% overall accuracy) and All-Level Guidance (39.40% overall accuracy), the CE + DisCL gets very limited improvement.

**Questions:**

In the Table 2 and Table 3, can you provide the results of CE + Text-only Guidance and CE + All-Level Guidance.

---

> ### Author Response · Authors · 2024-11-15
>
> Thank you for your valuable feedback! We have carefully reviewed your comments and address them as follows.
>
> > Existing works related to image guidance and contribution.
>
> One of our contributions is generating a complete and smooth spectrum of synthetic-to-real data and using this spectrum in training simultaneously. Different data are selected to be used at each epoch. Previous work, such as SyntheticData [1], introduces image guidance to control the similarity between generated images and the original input. However, although they incorporate image guidance in the data generation process (Real Guidance (RG) Strategy in [1]), each training run only utilizes a single image guidance level to produce synthetic images, rather than leveraging multiple guidance levels across training.
> In [1], specific image guidance levels are chosen for different few-shot settings, and the corresponding synthetic data is used exclusively for each setting. In contrast, our approach applies the entire spectrum of image guidance levels in a unified training process, offering a richer and more continuous range of synthetic-to-real features for curriculum designs and model adaptation.
>
> - [1] He, Ruifei, et al. "Is synthetic data from generative models ready for image recognition?." arXiv preprint arXiv:2210.07574 (2022).
>
> > Questions related to current results.
>
> DisCL primarily targets hard samples within the original dataset, focusing on generating synthetic data specifically for underrepresented classes, labeled as “Few” classes in Table 1. As shown in Table 1, the accuracy for “Few” classes improves from 17.90% (Text-only) and 19.17% (All-Level) to 23.64% (DisCL), highlighting the effectiveness of the curriculum paradigm applied in our method.
>
> However, given that only 136 out of 1,000 classes (1,643 out of 115,846 samples) fall into the “Few” category, the overall improvement in accuracy appears less pronounced, as the impact is more localized to these underrepresented classes.
>
> > Ablation studies on CIFAR100-LT and iNaturalist2018.
>
> We use ImageNet-LT as the primary dataset for our long-tail classification tasks and conduct ablation studies on this dataset. We plan to run additional experiments on CIFAR100-LT and iNaturalist2018, to further validate our approach and strengthen the robustness of our findings.

---

### Official Review · Reviewer_2cVr · 2024-11-08

**Soundness:** 3
**Presentation:** 3
**Contribution:** 2
**Rating:** 5
**Confidence:** 4

**Summary:**

The paper proposes a curriculum learning (CL) method that leverage synthetic image generation as data augmentation in combination with CL algorithm to tackle data-challenging tasks, e.g., long-tailed and low-quality distribution learning. The proposed method, DisCL, generattes images using both text and real images conditioning, with various image guidance scales to regulate the similarity to the real image, allowing for control over the hardness/complexity of the generated sample. CL is applied to select which complexity of samples (image guidance scale) to use based on the task at hand, e.g., for long tail learning an diverse-to-specific CL algorithm is used, while for low-quality image learning an adaptive algorithm is used. A first set of experiments compare DisCL versus baselines using data augmentation or balanced softmax for long tailed classification, showing positive impact mostly on less represented classes. A second set of experiments, test DisCL in the task of low-quality images using the iWildCam dataset. Here, DisCL is plugged to state-of-the-art fine-tuning techniques to show improved performance on both out-of-distribution and in-distribution examples.

**Strengths:**

- The method is simple and seems to work for both long-tailed and low-quality image classification.

**Weaknesses:**

1. Lacking of some relevant related works. The method does not mention nor compare against methods that already tried to leverage synthetic data as data augmentation to cope with unbalanced data, e.g., Hemmat-Askari et al 2023, or for representation learning / classification e.g., Tian et al 2024a and 2024b, Astolfi et al 2023. In particular, Hemmat-Askari et al 2023 seems quite related as they target the same task and use similar synthetic data generation approach while having some sort of adaptive curriculum learning (feedback guidance) which regulates the type of generation needed by the model. It would be nice to understands how DisCL compare against it. Finally, ALIA ( Dunlap et al 2023) is mentioned as a related work and a baseline in 3.1.2, but never presents in the results.

2. Weak / unclear experimental settings:
    - Resnet-10 choice is motivated for the comparison with LDMLR; However, most of the comparisons are with CUDA, which uses resent-32 for CIFAR-100 and resenet-50 for ImageNet. Do you expect you results to hold with these larger resnet?
    - Some experimental details are unclear to me.
        - it is not clear to me whether baselines and DisCL are trained for the same amount of iterations/epochs.
        - In the training details the authors say: _"To preserve a constant imbalance-ratio throughout all training stages and experiments, we undersample the non-tail samples at "each stage" so that ratio of tail-samples to non-tail samples matches the proportion of tail classes to non-tail classes present in the original data (13.6%)."_. If I am reading this correctly the authors say that they prefer to maintain imbalanced the dataset, despite having the possibility to rebalancing it with synthetic data. Why this choice?
        - The results on ImageNet-LT show small improvements w.r.t. to balanced softmax (BS) baseline (+1.5%). By looking at Hemmat-Askari et al 2023 results, the BS baseline is outperformed by a large marging. I understand that the number of generated data on Hemmat-Askari et al is on another scale (1.3M vs. 25K). Do you think the scale is enough to justify this difference?
        - Also, combining BS with DisCL sometimes leads to lower results than CE + DisCL (see Table 2). Is there any intuition why BS does seem to be as effective for DisCL
        - Bolding in table 2 is inconsistent


*_Tian, Yonglong, et al. "Stablerep: Synthetic images from text-to-image models make strong visual representation learners." Advances in Neural Information Processing Systems 36 (2024)._
* _Tian, Yonglong, et al. "Learning vision from models rivals learning vision from data." Proceedings of the IEEE/CVF Conference on Computer Vision and Pattern Recognition. 2024._
* _Hemmat, Reyhane Askari, et al. "Feedback-guided data synthesis for imbalanced classification." arXiv preprint arXiv:2310.00158 (2023)._
* _Astolfi, Pietro, et al. "Instance-conditioned gan data augmentation for representation learning." arXiv preprint arXiv:2303.09677 (2023)._
* _Dunlap, Lisa, et al. "Diversify your vision datasets with automatic diffusion-based augmentation." Advances in neural information processing systems 36 (2023): 79024-79034._

**Questions:**

Please respond to the weaknesses listed. Given the many clarification required I consider this work below the acceptance bar.

---

> ### Author Response · Authors · 2024-11-15
>
> Thank you for your detailed feedback! We carefully address your comments as below.
>
> > Relevant related works regarding data augmentation.
>
> Thank you for pointing out these related works! We would like to emphasize key distinctions between our approach and these methods.
> Our DisCL framework consists of two key components: generation of **spectrum of sync-to-real** data and the **curriculum learning paradigm** designed to effectively leverage this synthetic data. In contrast, SynCLR (Tian et al., 2024a; Tian et al., 2024b), DAIC-GAN (Astolfi et al., 2023), and Feedback-guided synthesis (Hemmat-Askari et al., 2023) primarily focus on generating data for downstream tasks without integrating curriculum learning.
> Specifically, feedback-guided data synthesis proposes to use the Feedback Criteria (Loss / Entropy / Hardness) from the pre-trained model to generate the synthetic data. However, once the data is generated, all synthetic samples are used indiscriminately in training without any selection or progressive filtering.
>
> In DisCL, beyond data generation with various image guidance, we further implement a curriculum learning paradigm to select data at each training epoch. This iterative selection helps the classifier incrementally bridge the gap between synthetic and real data distributions, enhancing model robustness and generalization.
>
> The ALIA result is shown as below. we will update Table 5 in our paper with these results.
>
> |  | OOD F1 Score | ID F1 Score |
> | --- | --- | --- |
> | ALIA | 36.9 (0.3) | 52.6 (0.4) |
> | DisCL | **38.2 (0.5)**  | **54.3 (1.4)**  |
>
> For experiments with larger models (ResNet32 and ResNet50), we expect our result will still hold. We will run them and use the results to strengthen our paper.
>
>
> > Questions related to training settings.
>
> For question related to training epochs, in our experiments, we always keep the total training epochs of DisCL less or equal to that of the baselines.
>
> For question about balanced ratio, DisCL is designed to improve performance on hard samples in the original dataset, specifically targeting “Few” classes in long-tail classification scenarios. In the data generation process, we focus exclusively on generating synthetic data for underrepresented classes, excluding “Medium” classes. Furthermore, generating synthetic data to achieve a fully balanced dataset would be both time- and resource-intensive. Thus, we maintain the original imbalance ratio in the generated dataset, allowing us to evaluate DisCL’s performance without changing population bias due to class distribution changes. To further validate our approach, we plan to conduct additional experiments using balanced synthetic datasets, aiming to strengthen our findings.
>
> > Questions related to current results compared with Hemmat-Askari et al 2023.
>
> For the difference between Feedback-guided synthesis (Hemmat-Askari et al., 2023) and DisCL, one key difference is the scale of the synthetic dataset used (1.3M samples vs. 25k in DisCL). Additionally, Feedback-guided synthesis generates synthetic data to augment the entire real dataset, whereas DisCL focuses on enhancing model performance specifically on challenging samples within the original dataset. For long-tail classification tasks, DisCL selectively generates synthetic data only for underrepresented classes (few classes in the original dataset). To further investigate this distinction, we plan to conduct additional experiments that control for data scale and class distribution.
>
> > Questions related to Table 2 results.
>
> Thank you for your questions and pointing out! We will correct our Table 2 as follows.
> | Imbalance Ratio=100 | Many | Medium | Few | Overall |
> | --- | --- | --- | --- | --- |
> | CE | 52.86 | 25.34 | 5.49 | 29.02 |
> | CE + CUDA | **54.55** | **26.07** | 5.43 | 29.85 |
> | CE + DisCL | 53.14 | 25.52 | **10.65** | **30.91** |
> | BS | 47.87 | 30.07 | 14.41 |31.61 |
> | BS + CUDA | 48.01 | **32.79** | 15.55 | 33.02 |
> | BS + DisCL | **49.02** | 29.02 | **19.07** | **33.08** |
>
> Since the Balanced Softmax (BS) function helps mitigate bias between dominant and underrepresented classes, BS+DisCL still achieves better performance than CE+DisCL, as DisCL continues to work with an imbalanced training dataset.

---

### Note · Authors · 2024-11-15

I have read and agree with the venue's withdrawal policy on behalf of myself and my co-authors.